# iNKT cells coordinate immune pathways to enable engraftment in nonconditioned hosts

Nicholas J Hess[1],* , Nikhila S Bharadwaj[1],* , Elizabeth A Bobeck[2], Courtney E McDougal[1], Shidong Ma[3], John-Demian Sauer[1] , Amy W Hudson[4] , Jenny E Gumperz[1]

Invariant natural killer T (iNKT) cells are a conserved population of innate T lymphocytes that interact with key antigen-presenting cells to modulate adaptive T-cell responses in ways that can either promote protective immunity, or limit pathological immune activation. Understanding the immunological networks engaged by iNKT cells to mediate these opposing functions is a key pre-requisite to effectively using iNKT cells for therapeutic applications. Using a human umbilical cord blood xenotransplantation model, we show here that co-transplanted allogeneic CD4$^+$ iNKT cells interact with monocytes and T cells in the graft to coordinate pro-hematopoietic and immunoregulatory pathways. The nexus of iNKT cells, monocytes, and cord blood T cells led to the release of cytokines (IL-3, GM-CSF) that enhance hematopoietic stem and progenitor cell activity, and concurrently induced PGE$_2$-mediated suppression of T-cell inflammatory responses that limit hematopoietic stem and progenitor cell engraftment. This resulted in successful long-term hematopoietic engraftment without pretransplant conditioning, including multi-lineage human chimerism and colonization of the spleen by antibody-producing human B cells. These results highlight the potential for using iNKT cellular immunotherapy to improve rates of hematopoietic engraftment independently of pretransplant conditioning.

## Introduction

Hematopoietic stem cell transplantation (HSCT) is used clinically to treat a variety of malignancies and nonmalignant hematologic diseases such as immunodeficiency disorders (Juric et al, 2016). Despite advances in pretransplant conditioning regimens, which are used to reduce tumor burden and ablate host immune cells to facilitate engraftment of transplanted hematopoietic stem and progenitor cells (HSPCs), adverse outcomes including graft failure, graft-versus-host disease (GVHD), and cancer relapse or progression remain highly problematic (Bishop et al, 2011; Tsai et al, 2016;

Chen et al, 2017; Yang et al, 2017). Thus, there is an ongoing need for new approaches to improve engraftment success, while minimizing GVHD and maintaining graft-versus-tumor (GVT) activity.

Invariant natural killer T (iNKT) cells are innate T lymphocytes that appear to play a constitutive role in promoting hematopoietic activity because iNKT cell–deficient mice show impaired hematopoiesis (Kotsianidis et al, 2006). In addition, iNKT cells may play important roles in regulating the outcome of immune reconstitution after human HSCT because they are one of the earliest T-cell subsets to reconstitute (Beziat et al, 2010; Dekker et al, 2020), and high frequencies of iNKT cells in graft tissue and at early times post-transplantation are associated with reduced GVHD (Haraguchi et al, 2004; Chaidos et al, 2012), as well as with lower rates of cancer relapse (de Lalla et al, 2011; Casorati et al, 2012). Prior studies have focused on the therapeutic potential of iNKT cells for preventing GVHD and promoting GVT after HSCT (Schneidawind et al, 2013; Mavers et al, 2017; Negrin, 2019). However, their therapeutic potential for promoting engraftment success remains poorly understood.

Importantly, iNKT cells are ideal candidates for use in allogeneic cellular immunotherapy applications such as HSCT. In contrast to MHC-restricted T cells that recognize foreign peptides, iNKT cells are restricted by CD1d molecules, which are highly conserved antigen-presenting molecules that present lipids and glycolipids (Brigl & Brenner, 2004). Because of the limited polymorphism of human CD1d genes (Han et al, 1999), iNKT cells do not mediate allo-responses to APCs from unrelated individuals. Moreover, iNKT cells are activated by multiple endogenous pathways including recognition of self-lipids presented by CD1d, TCR-independent cytokine exposure, and integrin-mediated signaling (Brigl et al, 2003, 2011; Fox et al, 2009; Sharma et al, 2018). Thus, they can constitutively perform immunological functions after transplantation, without requiring pharmacological activation.

Here, we have used a model of human umbilical cord blood (UCB) transplantation into immunodeficient NOD/SCID/$\gamma_c{}^{-/-}$ (NSG) mice to investigate how co-transplantation of allogeneic human CD4$^+$ iNKT cells affects engraftment outcomes. In clinical settings,

[1]Department of Medical Microbiology and Immunology, University of Wisconsin School of Medicine and Public Health, Madison, WI, USA    [2]Department of Animal Science, 201F Kildee Hall, Iowa State University, Ames, IA, USA    [3]QLB Biotherapeutics, Inc., Boston, MA, USA    [4]Department of Microbiology and Immunology, Medical College of Wisconsin, Milwaukee, WI, USA

Correspondence: jegumperz@wisc.edu
*Nicholas J Hess and Nikhila S Bharadwaj contributed equally to this work

UCB transplantation is associated with low incidence of GVHD (Keating et al, 2019), but it is limited by high rates of graft failure (about 20% of transplant cases), and by longer median times to neutrophil and platelet recovery than bone marrow or G-CSF–mobilized peripheral blood transplants (Lucchini et al, 2015; Tsai et al, 2016). A central component of the graft failure and delayed immune reconstitution in UCB transplantation is thought to be the comparatively low numbers of HSPCs in UCB samples (Brown & Boussiotis, 2008). In addition, in-flammation, triggered by conditioning-associated damage to the host and/or by transplantation-associated activation of donor-derived im-mune cells, likely plays an important role in UCB graft failure (Ramadan & Paczesny, 2015; Luis et al, 2016). Using nonconditioned mice to minimize damage-associated inflammation in the host, we observed that co-transplanted allogeneic human iNKT cells orchestrated a co-ordinated response involving the secretion of cytokines that promote hematopoietic activity, while concurrently regulating the production of inflammatory cytokines that adversely affect engraftment.

## Results

### Human hematopoietic engraftment in nonconditioned hosts

Pretransplant conditioning is typically required for successful engraftment of purified human HSPCs transplanted into immu-nodeficient murine hosts (Waskow et al, 2009; McIntosh et al, 2015; Sippel et al, 2019). However, prior studies have established that immunodeficient mice with mutations that lead to reduced func-tioning of *c-kit* (the receptor for stem cell factor) support robust human hematopoietic engraftment without requiring pretransplant conditioning (Cosgun et al, 2014; McIntosh et al, 2015; Yurino et al, 2016; Czechowicz et al, 2019; Hess et al, 2020b), presumably because the transplanted human cells (which have normal *c-kit* activity) have a competitive advantage over the endogenous murine he-matopoietic cells. Therefore, we used NBSGW mice, which bear a hypomorphic $W^{41J}$ *c-kit* mutation as a positive control to gen-erate successful human hematopoietic engraftment in a non-conditioned host. 3 mo after transplanting NBSGW mice with purified human CD34$^+$ HSPCs, a sizeable population of human immune cells was present in the murine bone marrow (Fig 1A, top row, left panel). The human immune compartment contained a CD34$^+$ subset comprising multiple sub-populations, including multi-potent progenitors (CD38$^-$CD45RA$^-$), committed progenitors (CD38$^+$CD45RA$^+$), myeloid pro-genitors (CD33$^+$ or CD123$^+$), and early B lymphocytic lineage cells (CD19$^+$) (Fig 1A, top row, middle and right panels). These results indicated that consistent with prior analyses (Cosgun et al, 2014; McIntosh et al, 2015; Yurino et al, 2016; Czechowicz et al, 2019; Hess et al, 2020b), transplanting human cord blood HSPCs into nonconditioned NBSGW mice resulted in successful hematopoietic engraftment that was sustained for at least 3 mo.

We next investigated transplantation of total human umbilical cord blood mononuclear cells (CBMCs) into nonconditioned NSG mice (a strain that is similarly immunodeficient as NBSGW, but is wild-type for *c-kit*). When we transplanted CBMCs alone (5 × 10$^6$ cells per mouse), there was typically only a small population of human cells detected in the bone marrow after 3 mo (Fig 1A, middle

row). The human population found in these mice showed little or no positive staining for CD34 (Fig 1A, middle row), and instead the human population comprised almost exclusively the T cells (Fig 2C, middle panel), indicating hematopoietic failure had occurred. In contrast, when we co-transplanted CBMCs (5 × 10$^6$ cells) along with allogeneic CD4$^+$ iNKT cells (1 × 10$^6$ cells) into nonconditioned NSG mice we observed abundant human cells in the murine bone marrow after 3 mo (Fig 1A, bottom row). Similar to what we had observed in HSPC-transplanted NBSGW mice, the human compartment in the bone marrow of NSG mice co-transplanted with human CBMCs + iNKT cells contained a CD34$^+$ subset made up of multiple progenitor sub-populations (Fig 1A, bottom row). Quantitative flow cytometric analysis confirmed that mice that received CBMCs + iNKT cells had significantly higher numbers of human CD34$^+$ cells in the murine bone marrow than mice that received CBMCs alone (Fig 1B), sug-gesting that co-transplantation of iNKT cells promoted human cord blood hematopoietic engraftment.

Because most of the human cells in the murine bone marrow at 3 mo posttransplantation were negative for CD34 (Fig 1A), we performed flow cytometric analyses to identify types of differentiated cells. As expected based on prior reports indicating that myelopoiesis is comparatively efficient in Kit-mutant mice (Cosgun et al, 2014; McIntosh et al, 2015; Hess et al, 2020b), we observed populations of CD34-negative human cells expressing the myeloid lineage markers CD33 or CD123 in NBSGW mice transplanted with purified cord blood HSPCs (Fig 2A, left panel). Similar CD33$^+$ or CD123$^+$ populations were also observed in the bone marrow of NSG mice transplanted with CBMCs + iNKT cells (Fig 2A, right panel), whereas these cells were absent in the bone marrow of NSG mice transplanted with CBMCs alone (Fig 2A, middle). Human cells expressing B-lineage markers (CD19 and CD38) were also present in both the HSPC-transplanted NBSGW mice and CBMC + iNKT-transplanted NSG mice, but were rare or absent in NSG mice that received CBMCs alone (Fig 2B). Staining for T lymphocytes and NK cells revealed that nearly all the human cells in the bone marrow of NSG mice that received CBMCs alone were T cells (Fig 2C, middle), whereas the bone marrow of HSPC-transplanted NBSGW mice and CBMC + iNKT-transplanted NSG mice contained few T cells (Fig 2C, left and right panels). Phenotypic analysis of the T cells present in the bone marrow of mice transplanted with CBMCs alone compared with CBMCs + iNKT cells showed that they did not differ significantly in their frequencies of CD4$^+$ or CD8$^+$ T cells, but the T cells of mice that received CBMCs alone appeared strongly biased towards an activated effector phenotype, with significantly higher frequencies of CD45RO$^+$ cells and lower frequencies of CD45RA$^+$ and a higher proportion of T cells showing a blasting phenotype (Fig 2D). Because we have previously established that human CD3$^+$ T cells do not arise from HSPCs under the experimental conditions used here (Hess et al, 2020a, 2020b), these results suggest that in mice that received CBMCs alone, T cells that were present in the starting graft tissue became activated into an effector status and this was associated with a failure of multi-lineage immune reconstitution. In contrast, in mice that received CBMCs + iNKT cells, T-cell activation was more limited and there was successful immune recon-stitution of myeloid and B-lineage cells at 3 mo posttransplant.

### Productive B-cell engraftment

We noted that human B-cell chimerism appeared more robust in the bone marrow of NSG mice transplanted with CBMCs + iNKT cells

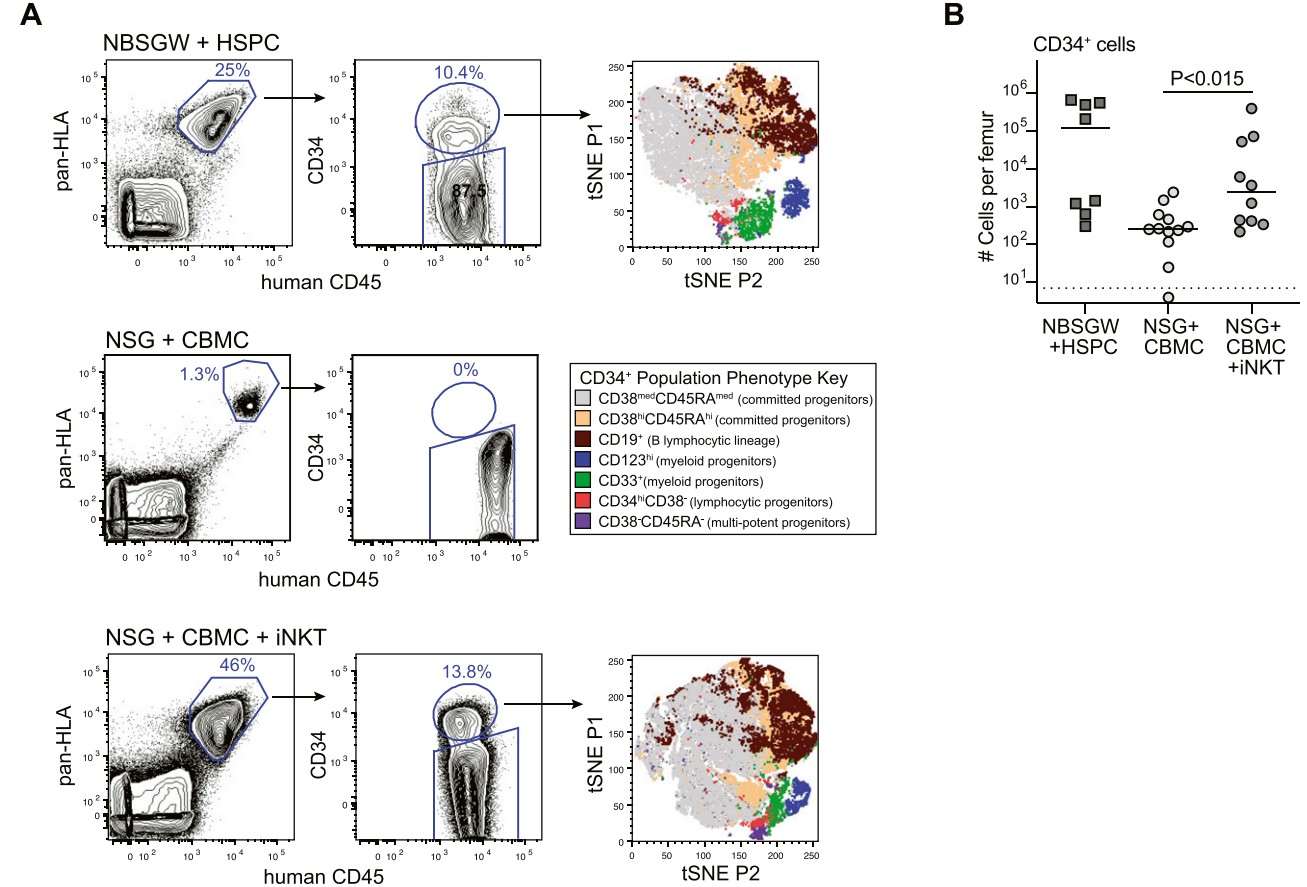

**Figure 1. Co-transplanting nonconditioned NSG mice with cord blood mononuclear cells (CBMCs) and invariant natural killer T cells phenocopies HSPC engraftment observed in a KIT-deficient mouse strain.**

**(A)** Flow cytometric analysis to detect human cells expressing CD34 in murine bone marrow at 3 mo posttransplantation. Top row shows results from an NBSGW (KIT deficient) mouse transplanted with $5 \times 10^5$ purified human cord blood CD34$^+$ HSPCs; (t-SNE) t-distributed stochastic neighbor embedding plot was generated from a concatenation of the human CD34$^+$ subsets of four similarly transplanted NBSGW mice (see color-coded key for populations in middle row). Middle row shows results from an NSG mouse transplanted with $5 \times 10^6$ human CBMCs. Bottom row shows results from an NSG mouse transplanted with $5 \times 10^6$ human CBMCs and $0.5 \times 10^6$ allogeneic CD4$^+$ invariant natural killer T cells; tSNE plot was generated from a concatenation of the human CD34$^+$ subsets of three similarly transplanted NSG mice. **(B)** Aggregated data showing numbers of human CD34$^+$ cells detected in bone marrow at 3 mo posttransplantation. Each symbol shows the result from one femur bone of an individual mouse, with bars showing the median; symbol below dotted line indicates specifically stained cells were not detected. Data are aggregated from six independent experiments; *P*-value calculated using a two-tailed Mann–Whitney test.

compared with NBSGW mice transplanted with purified UCB HSPCs (Fig 3A). We therefore delved further into the characteristics of human B-cell reconstitution in NSG mice transplanted with CBMCs + iNKT cells. There was little or no evidence of human B cells in the murine bone marrow for the first 4–5 wk after transplantation, and a population of human cells staining brightly for CD38 and co-expressing CD19 emerged after ~6 wk (Fig 3B). Notably, this is similar to the timing observed for the emergence of B cells after hematopoietic transplantation in humans (van der Maas et al, 2019).

Human B-cell chimerism was maintained in the NSG bone marrow and spleen for at least 9 mo after transplantation of CBMCs + iNKT cells (Fig 3C). The human compartment in the spleens of mice transplanted with CBMCs + iNKT cells appeared dominated by B cells, but we also detected human monocytic cells (CD14$^+$) and human T cells (CD3$^+$) that remained for at least 6 mo after trans-plantation (Fig 3D). We also performed histological analyses of spleen tissue from NSG mice transplanted with CBMCs + iNKT

cells at 10-mo posttransplantation. Immunohistochemical staining revealed large aggregates of CD20$^+$ cells (a marker of mature B cells), and within these areas were isolated cells staining positively for Bcl-6, which is a marker of germinal centers (Fig 3E, left panel). In addition, isolated cells staining strongly for human IgM or human IgG were visible in these areas (Fig 3E, middle and right panels), suggesting the presence of B cells expressing successfully rearranged surface immunoglobulin. Plasma samples taken at 5–9 mo post-transplantation typically contained clearly detectable levels of human immunoglobulin, confirming the presence of immunoglobulin-producing human B cells (Fig 3F). Thus, nonconditioned NSG mice given CBMCs + iNKT cells developed a durable and productive human B-cell compartment.

To confirm that the enhanced human immune reconstitution observed in NSG mice given CBMCs + iNKT cells was selectively associated with iNKT cells and not simply due to co-transplantation of cultured human T cells, we tested mice that were given CBMCs

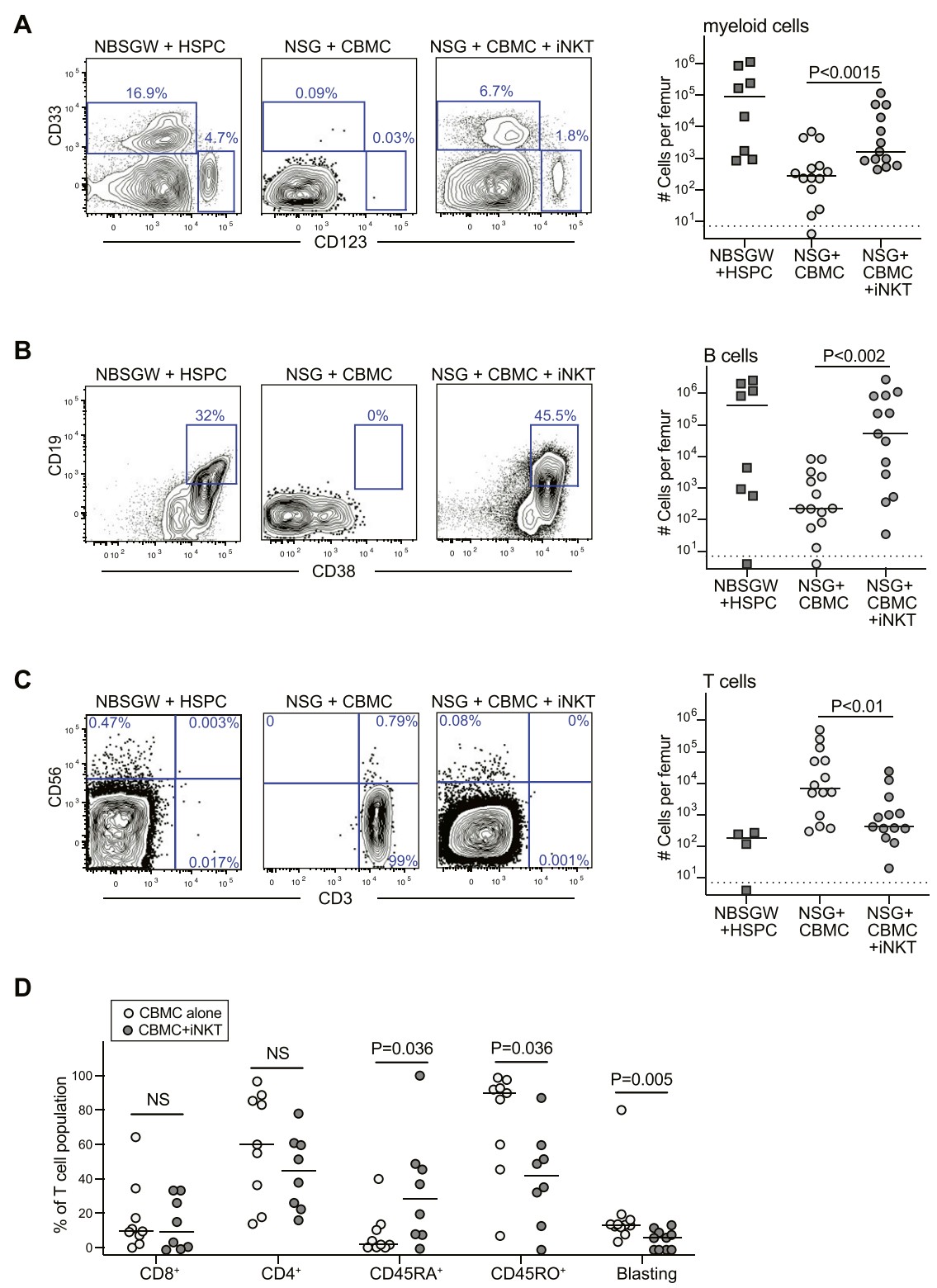

**Figure 2. Establishment of multi-lineage immune engraftment.**
**(A)** Flow cytometric analysis of murine bone marrow samples collected at 3 mo posttransplantation showing staining of the CD34-negative human population for myeloid lineage markers (CD33 and CD123). Plot on right shows numbers of human myeloid cells detected in one femur bone of mice in the indicated treatment groups, with bars showing the median for each group (symbol below dotted line indicates no specifically detectable cells). Results aggregated from nine independent experiments; P-values calculated using a two-tailed Mann–Whitney test. **(B)** Staining for B-lineage cells, and aggregated data for numbers of B cells in bone marrow. **(C)** Staining for T cells and NK cells, and aggregated data for numbers of T cells in bone marrow. **(D)** Aggregated data for phenotypic analysis of T cells detected in bone marrow of mice that received cord blood mononuclear cells alone (light symbols) compared with cord blood mononuclear cells + invariant natural killer T cells (dark symbols).

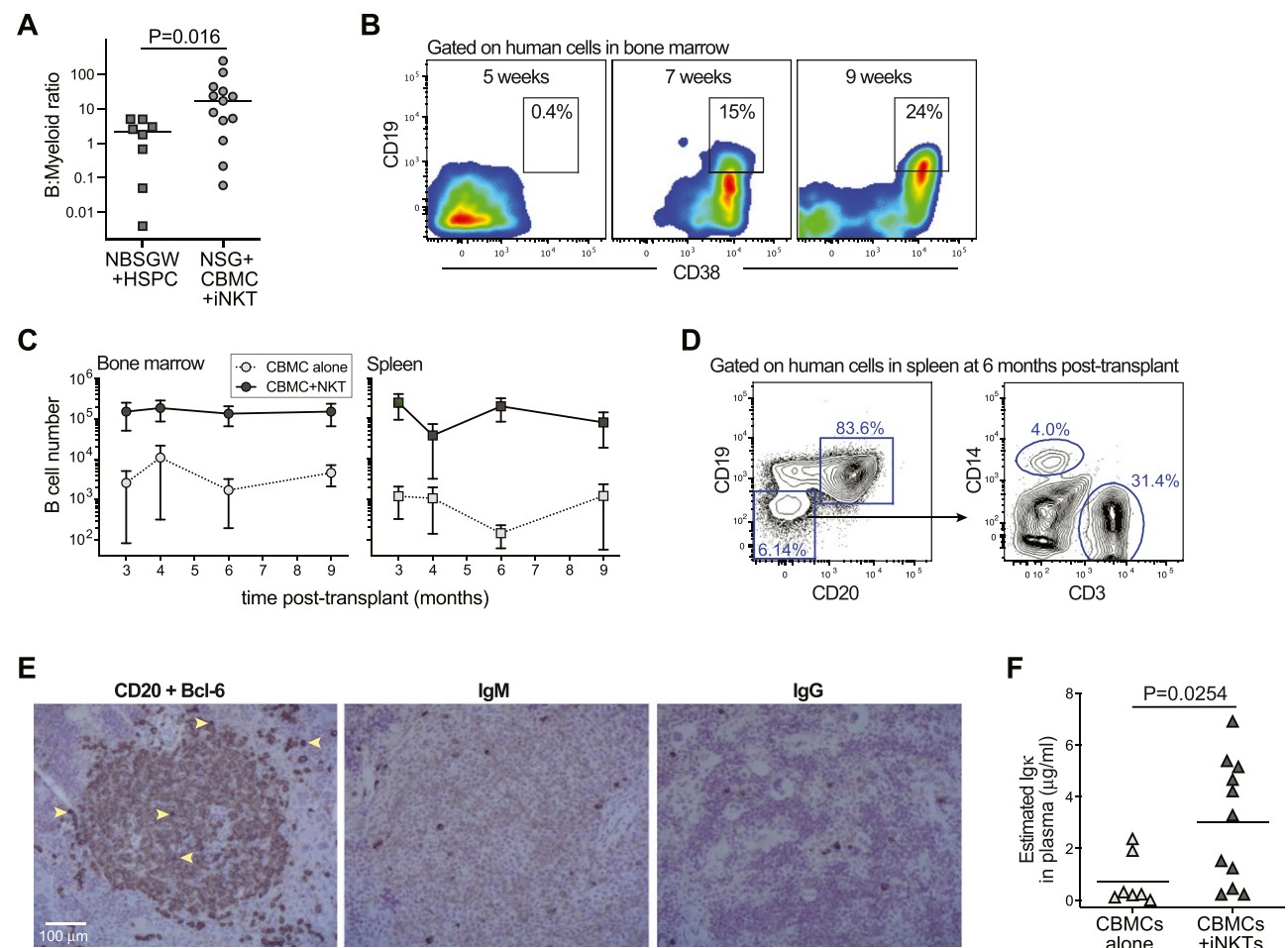

**Figure 3. Establishment of a productive human B-cell compartment in NSG mice given cord blood mononuclear cells (CBMCs) and invariant natural killer T (iNKT) cells.**
**(A)** Human compartment in bone marrow of NSG mice transplanted with CBMCs and iNKT cells show significantly greater ratio of B cells to myeloid lineage cells than in NBSGW mice transplanted with purified CD34$^+$ HSPCs. **(B)** Flow cytometric analysis of human cells in bone marrow at the indicated time points posttransplantation showing the emergence of B-lineage cells. **(C)** Mean (±SEM) numbers of human B cells detected in murine bone marrow (left plot) or spleen (right plot) at the indicated times postengraftment. Data are aggregated from six independent experiments; with each symbol representing the average of three to eight mice. **(D)** Flow cytometric analysis of human cells in spleen of an NSG mouse given CBMCs and iNKTs at 6 mo posttransplantation. **(E)** Immunohistochemical analysis of serial tissue sections from the spleen of a mouse transplanted 10 mo earlier with human CBMCs and iNKT cells. Left panel shows co-staining for human CD20 (brown color) and Bcl-6 (dark purple color; examples of Bcl-6$^+$ cells indicated by yellow arrows). Middle panel shows staining for human IgM ($\mu$ chain). Right panel shows staining for human IgG ($\gamma$ chain). Images from light microscopic analysis at 10× magnification. **(F)** Plasma samples were collected at 5–9 mo posttransplantation, and tested using an ELISA specific for human Igκ light chain. Total amounts of Igκ were estimated by comparing plasma sample titers to a pooled human AB serum standard, and multiplying by previously determined values for human immunoglobulin (Cassidy & Nordby, 1975). Each symbol represents the mean of three replicate analyses of a plasma sample from an individual mouse. Data aggregated from three independent experiments; *P*-value calculated by two-tailed Mann–Whitney test.

combined with allogeneic polyclonal CD4$^+$ T cells that were expanded in vitro under the same conditions as those used for the iNKT cells. These mice had too few human cells in the bone marrow for reliable analysis at 12 wk posttransplantation (data not shown), and at 10 wk posttransplantation they contained only a small human cell population that lacked B cells (Fig S1). We also found that human iNKT cells transplanted into nonconditioned NSG mice were detectable in the murine bone marrow within 2 d following transplantation and remained detectable there for at least 3 wk (Fig S2A). Flow cytometric analysis of mice transplanted with a 1:1 ratio of iNKT cells and CMBCs revealed that by 5 wk after transplantation, the iNKT cells comprised <1% of the human cells in bone marrow (Fig S2B), suggesting that the iNKT cells expanded less efficiently

than the cord blood T cells in the weeks after transplantation. Together, these results confirmed that iNKT cells promote immune reconstitution by allogeneic CBMC-derived cells.

## Role of accessory cells

We next investigated whether the impact of iNKT cells was due to direct interactions with HSPCs. Co-transplanting iNKT cells with purified HSPCs was not sufficient to promote successful engraftment (Fig 4A), suggesting other cell types present in CBMC grafts were required. Using a live cell in vitro imaging system, we investigated the amount of cell–cell interaction between fluorescently labeled allogeneic iNKT cells and each of the three most

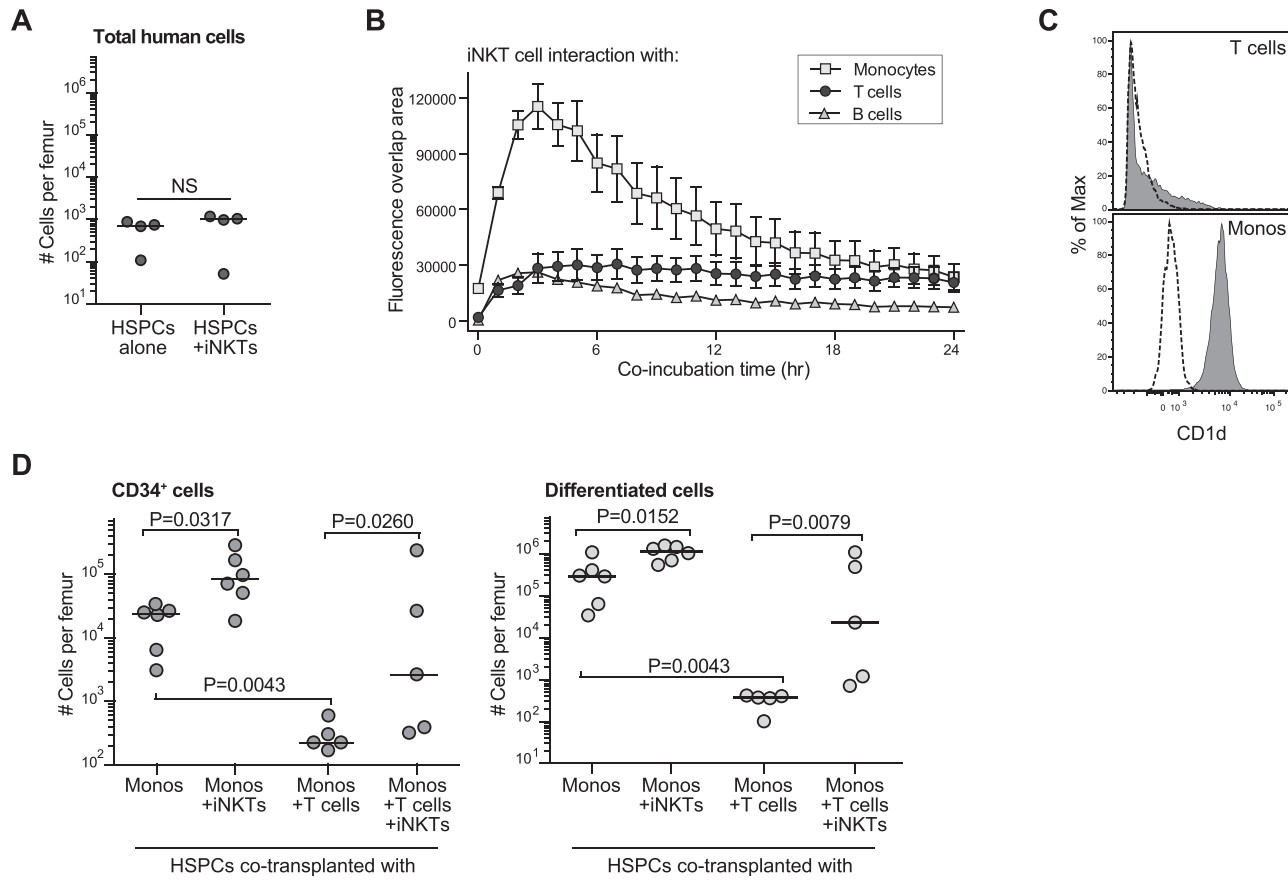

**Figure 4. Invariant natural killer T (iNKT) cells interact with cord monocytes to promote HSPC engraftment.**
**(A)** Analysis of the number of total human cells detected in one femur of NSG mice transplanted with 0.5–1 × 10⁵ purified CD34⁺ cells alone or in combination with 1 × 10⁶ allogeneic CD4⁺ iNKT cells. **(B)** Analysis of the amount of cell–cell interaction between iNKT cells and the three most abundant cell types found in cord blood. iNKT cells were labeled with a red dye and placed into tissue culture wells with equivalent numbers of purified cord blood monocytes, T cells, or B cells that had been labeled with a green dye. Fluorescence microscopic images were taken at 20× magnification every 60 min using an IncuCyte Live Cell imaging system. The plot shows the mean ± SEM area of fluorescence overlap calculated from three replicate wells for each co-culture condition. **(C)** Flow cytometric staining of CD1d (filled histograms) compared with isotype (dotted lines) on cord blood T cells (upper panel) or monocytes (lower panel). **(D)** NSG mice were transplanted with 0.5–1 × 10⁵ purified CD34⁺ cells in combination with the indicated cell types. Left plot shows number of human CD34⁺ cells per femur, and right plot shows number of lineage⁺ human cells (excluding T cells) per femur at 3 mo posttransplantation. Results aggregated from three independent experiments; P-values calculated using a two-tailed Mann–Whitney test.

abundant immune populations present in the CBMC grafts (T cells, monocytes, and B cells). We observed the highest amount of fluorescence signal overlap between iNKT cells and monocytes, and the lowest amount between iNKT cells and B cells, with slightly elevated overlap signal for iNKT cells and cord T cells (Fig 4B). Notably, although CD1d staining appeared uniformly positive on cord blood monocytes, CD1d appeared only to be expressed at low levels on a fraction of cord T cells (Fig 4C). Interestingly, although CD1d is also uniformly expressed at low cell surface levels by cord blood B cells (Sarvaria et al, 2016), we observed little evidence of sustained iNKT-B-cell contact.

Because these results suggested a potential role for the monocytes and/or T cells that are present in CBMC samples, we tested the impact of these cell types on HSPC engraftment in vivo. Purified HSPCs were co-transplanted with purified autologous monocytes, or purified autologous monocytes and purified autologous T cells, in the presence or absence of allogeneic iNKT cells. Co-transplantation of HSPCs with monocytes and iNKT cells resulted in significantly higher numbers of human CD34⁺ cells in the

murine bone marrow after 3 mo than in HSPCs transplanted only with autologous monocytes (Fig 4D, left plot). Similarly, significantly higher numbers of total differentiated cells (all human lineage⁺ cell types, excluding T cells) were detected in mice that received HSPCs + monocytes + iNKT cells (Fig 4D, right plot). This suggested that interactions between iNKT cells monocytes have pro-hematopoietic effects that promote HSPC engraftment and immune reconstitution in non-conditioned hosts.

In contrast, inclusion of autologous T cells with the HSPCs + monocytes resulted in significantly reduced numbers of CD34⁺ and lineage⁺ cells compared with mice that received only HSPCs + monocytes (Fig 4D), indicating that the presence of autologous T cells had an adverse impact on human HSPC engraftment. However, when allogeneic iNKT cells were co-transplanted with the monocytes and T cells, the engraftment of CD34⁺ cells was increased and enhanced reconstitution of non–T-cell lineage⁺ cells was observed (Fig 4D). Thus, the co-transplanted iNKT cells appeared to counteract the adverse effects of cord T cells on HSPC engraftment and immune reconstitution.

## Pro-hematopoietic activity of iNKT cells

To further investigate the pro-hematopoietic effects of co-transplanted iNKT cells, we assessed the ability of our iNKT cells to produce factors that promote hematopoietic expansion. Purified CD34$^+$ cells were cultured in serum-free medium containing a cytokine cocktail (stem cell factor [SCF], thrombopoietin [TPO], Flt-3L, and IL-7) with concentrations adjusted to support HSPC differentiation at a somewhat suboptimal level. In parallel, CD34$^+$ cells were cultured in the presence of transwell inserts containing iNKT cells alone or iNKT cells and anti-CD3/anti-CD28 coated beads. After 10 d in culture, the number of cells in the lower transwell expressing lineage markers (CD33, CD38, and CD45RA) was quantitated. Although the presence of iNKT cells alone in the upper transwell had no detectable impact, the differentiation of HSPCs in the lower well was markedly increased when CD3/CD28-activated iNKT cells were present in the upper transwell (Fig 5A). Thus, this analysis indicated that activated but not resting iNKT cells secreted factors that promote the hematopoietic activity of CD34$^+$ HSPCs.

In prior studies, we have observed that even weak TCR agonism resulting from exposure to CD1d molecules presenting cellular antigens was sufficient to activate human CD4$^+$ iNKT cells to produce GM-CSF (Wang et al, 2008). We also found that the *IL3* locus of resting human CD4$^+$ iNKT cells resembled that of *CSF2* (GM-CSF) in showing a histone acetylation pattern associated with active chromatin (Wang et al, 2012). Thus, we hypothesized that human CD4$^+$ iNKT cells may also be able to produce IL-3 in response to comparatively low levels of TCR stimulation. To test this, we assessed secretion of IL-3 versus GM-CSF by our iNKT cells in response to recombinant CD1d molecules loaded with titrated doses of the synthetic antigen $\alpha$-GalCer. Compared with GM-CSF, half maximal production of IL-3 required ~twofold lower doses of $\alpha$-GalCer (Fig 5B), indicating that iNKT cells produce IL-3 even more readily than GM-CSF in response to TCR stimulation.

Because these results suggested that exposure to CD1d$^+$ cells might be sufficient to activate iNKT cell secretion of both GM-CSF and IL-3, we assessed the levels of these cytokines in culture supernatants of iNKT cells with cord monocytes and T cells. Co-cultures of all three cell types consistently showed elevated levels of both GM-CSF and IL-3 (Fig 5C). Co-cultures of iNKT cells + monocytes or iNKT cells + T cells sometimes contained elevated levels of GM-CSF or IL-3, but overall did not reproducibly differ from iNKT cells alone, suggesting that the nexus of iNKT cells and cord monocytes with autologous T cells was key for inducing release of GM-CSF and IL-3 (Fig 5C). To confirm the presence of this iNKT cell-dependent cytokine pathway in vivo, we transplanted NSG mice with CBMCs alone or CBMCs + iNKTs then harvested bone marrow and blood plasma after 72 h and analyzed human GM-CSF and IL-3 by ELISA. Significantly elevated levels of GM-CSF and IL-3 were detected in bone marrow of mice that received CBMCs + iNKT cells compared with those that got only CBMCs, and these cytokines also appeared modestly elevated in the blood (Fig 5D).

We next investigated whether factors secreted during co-cultures of iNKT cells with cord monocytes and autologous T cells were sufficient to promote the hematopoietic activity of cord blood HSPCs. Culturing purified CD34$^+$ HSPCs in the presence of transwell inserts containing a combination of iNKT cells +

monocytes + autologous T cells led to significantly enhanced hematopoietic activity, whereas exposure to transwells containing any of these cells alone or in pairs did not increase the output of differentiated cells (Fig 5E). Together, these results demonstrate a pro-hematopoietic nexus consisting of iNKT cells, cord monocytes, and autologous T cells.

## Adverse effects of cord T cells

The observation that hematopoietic engraftment fared significantly worse in mice co-transplanted with purified CD34$^+$ HSPCs + autologous monocytes + T cells compared to mice that received HSPCs + autologous monocytes alone (Fig 4D) suggested that cord T cells promote graft failure in this model. Consistent with this, we found that NSG mice given T-depleted CBMCs showed significantly greater numbers of CD34$^+$ and lineage$^+$ cells (excluding T cells) in the murine bone marrow after 3 mo compared to those given total CBMCs (Fig 6A). In addition, we found that the in vitro expansion of HSPC-derived cells was significantly limited when they were co-cultured with CBMCs in the presence of anti-CD3 antibody stimulation, but expansion recovered when a blocking antibody against either IFN-$\gamma$ or TNF-$\alpha$ was included (Fig 6B). Analysis of blood plasma samples taken at regular intervals during the first 3 mo posttransplantation revealed that mice transplanted with CBMCs alone showed clearly detectable human IFN-$\gamma$ in the blood that peaked at 6 wk after transplantation (Fig 6C), suggesting that the cord T cells become activated after transplantation in this model. Mice given CBMCs + iNKT cells showed significantly lower plasma IFN-$\gamma$ concentrations at 3 wk posttransplantation ($P$ = 0.0033, n = 13) and also showed a trend towards reduced levels at later time points (Fig 6C). Thus, co-transplantation of iNKT cells was associated with reduced inflammatory activation after CBMC transplantation.

We have previously shown that human CD4$^+$ iNKT cells induce monocytes from adult peripheral blood to differentiate into cells that potently suppress adult T-cell IFN-$\gamma$ production (Hegde et al, 2009, 2011). We therefore hypothesized that iNKT cells might interact with monocytes from cord blood to silence the activation of cord T cells. Using an in vitro co-culture system, we found that cord T cells cultured with autologous monocytes and iNKT cells showed significantly reduced proliferation and produced significantly less TNF-$\alpha$ and IFN-$\gamma$ after PMA and ionomycin stimulation compared with cord T cells cultured only with autologous monocytes (Fig 6D). In addition, we confirmed that transplanting purified cord T cells with autologous monocytes was sufficient to produce detectable levels of human IFN-$\gamma$ in the plasma that peaked at 6 wk after transplantation into NSG mice, and co-transplantation of iNKT cells led to reduced circulating IFN-$\gamma$ (Fig 6E). Taken together, these results suggested that iNKT cells interact with cord monocytes to suppress the responses of cord T cells.

## Cord T-cell suppression–mediated via eicosanoid production

In prior analyses, we observed that CD4$^+$ iNKT cells activated multiple inhibitory pathways in human monocytes from adult peripheral blood, including ones that were mediated via cell contact (e.g., up-regulation of PD-L1 and PD-L2) as well as ones mediated by secreted factors (e.g., IL-10 production) (Hegde et al, 2009, 2011). We therefore tested whether contact with the iNKT cells

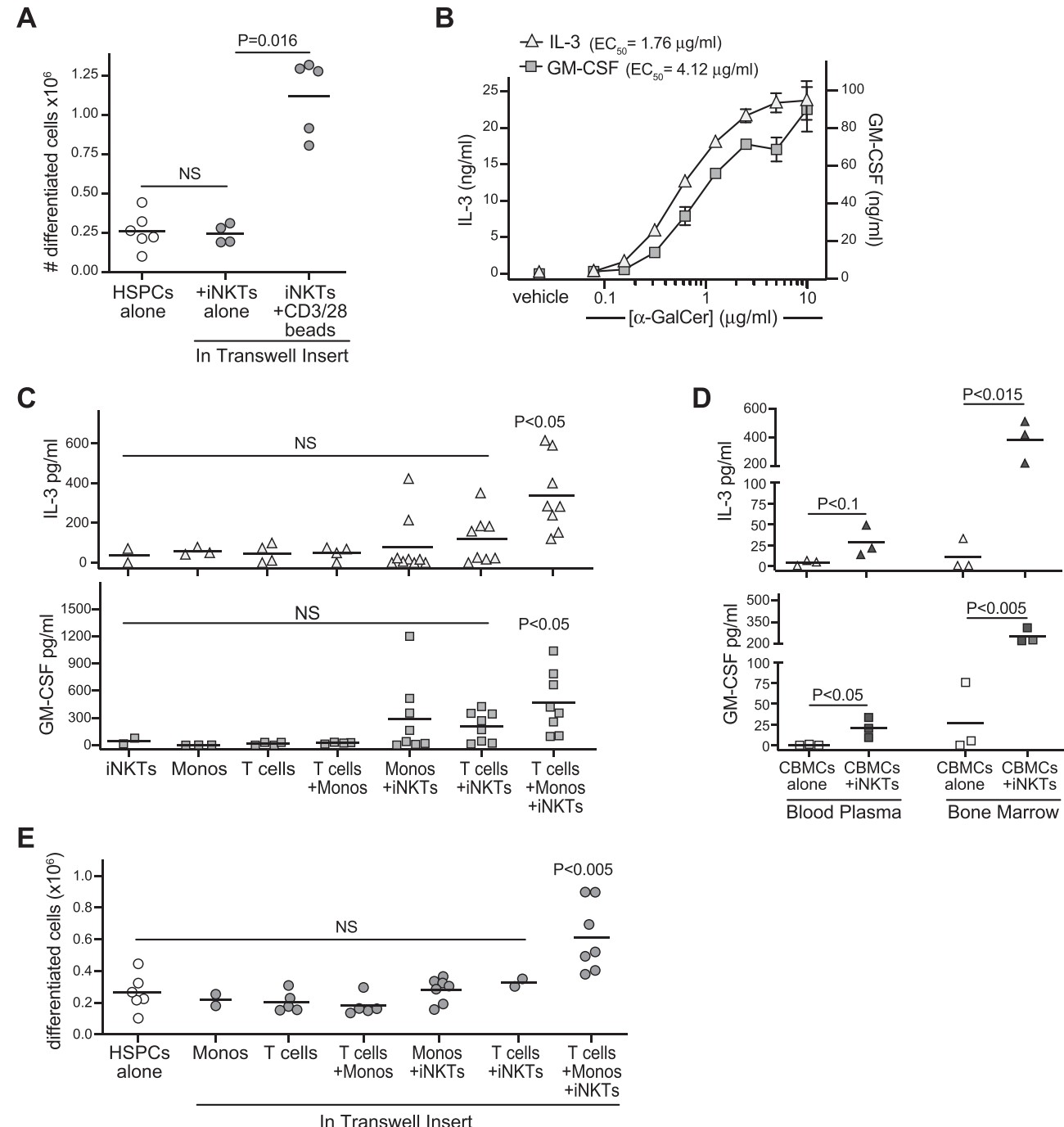

**Figure 5. Invariant natural killer T (iNKT) cells show pro-hematopoietic activity that is activated by exposure to cord T cells and monocytes.**
**(A)** Purified CD34+ HSPCs were seeded at 2.5 × 10^4 cells/well and cultured in the lower wells of transwell plates in medium containing a hematopoietic cytokine cocktail. Where indicated, iNKT cells were added to the transwell inserts alone or in combination with CD3/CD28 Dynabeads. After 10 d, the number of lineage+ cells in the lower transwell was quantitated. Plot shows aggregated results from six independent experiments with bars at the means and *P*-values calculated by a two-tailed Mann–Whitney test. **(B)** CD4+ iNKT cells were exposed to plate-bound recombinant CD1d–Fc fusion protein that had been pretreated with the indicated concentrations of *α*-GalCer lipid antigen or with vehicle alone. Plot shows amounts of IL-3 (left axis) or GM-CSF (right axis) detected in the culture supernatant after 24 h. Symbols show means of three replicate wells, with error bars showing standard deviations (not always large enough to be visible). **(C)** iNKT cells, isolated cord T cells, and/or autologous cord monocytes were cultured alone or in the indicated combinations for 48 h, and amounts of GM-CSF and IL-3 in the culture supernatants were quantitated. Each symbol represents the mean amount of cytokine detected from two to three replicate wells of an independent analysis, with bars showing means of aggregated results. *P*-values were calculated using a two-tailed Mann–Whitney test for each condition in comparison to iNKT cells alone. **(D)** Analysis of human GM-CSF and IL-3 levels in bone marrow or blood plasma of NSG mice transplanted with cord blood mononuclear cells alone or cord blood mononuclear cells + iNKTs. Each symbol represents the mean amount of cytokine detected from an individual mouse; *P*-values calculated using a two-tailed unpaired *t* test. **(E)** Purified CD34+ HSPCs were seeded at 2.5 × 10^4 cells/well and cultured in the lower wells of transwell plates in medium containing a hematopoietic cytokine cocktail. Where indicated, iNKT cells, cord monocytes, and/or autologous cord T cells were added to the transwell inserts. After 10 d, the number of lineage+ cells in the lower transwell was quantitated. Plot shows aggregated results from five independent experiments with bars at the means. *P*-values were calculated using a two-tailed Mann–Whitney test for each transwell culture condition in comparison to HSPCs cultured alone.

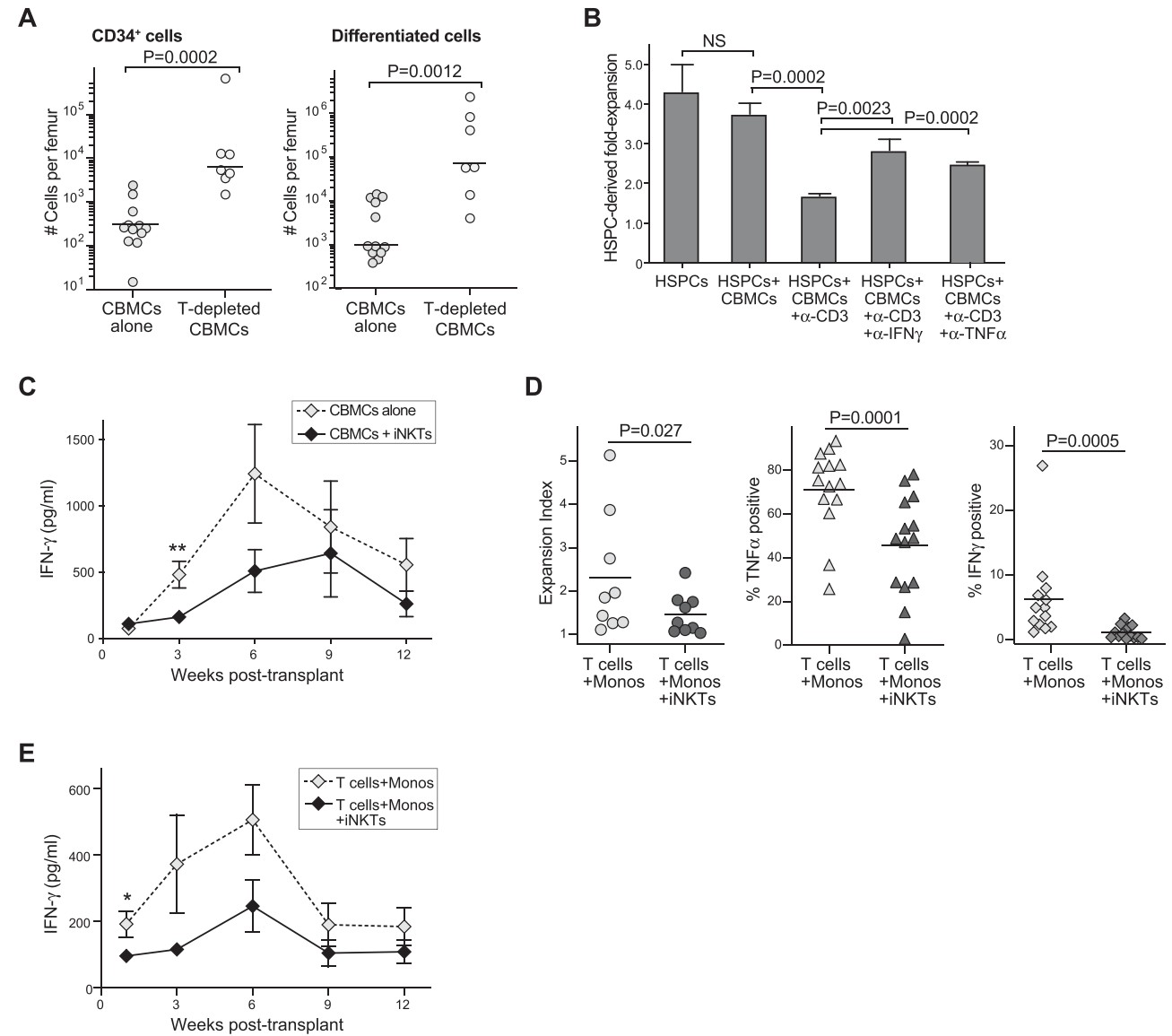

**Figure 6.  Invariant natural killer T (iNKT)–monocyte interactions inhibit cord T-cell responses.**
**(A)** NSG mice were transplanted with 5 × 10⁶ total cord blood mononuclear cells (CBMCs) or 3 × 10⁶ T-cell–depleted CBMCs. Left plot shows number of CD34⁺ human cells and right plot shows number of lineage⁺ human cells (excluding T cells) in one femur at 3 mo posttransplantation. Results aggregated from seven independent experiments; *P*-values calculated by two-tailed Mann–Whitney test. **(B)** Purified HLA-A2⁺ CD34⁺ HSPCs were cultured alone or in the presence of CD34-depleted HLA-A2⁻ CBMCs in medium containing a hematopoietic cytokine cocktail. Where indicated, anti-CD3, and anti-CD28 antibodies were added to activate the cord T cells, and anti-IFN-γ or anti–TNF-α–blocking antibodies were added. Plot shows the fold expansion of HLA-A2⁺ cells after 7 d (mean ± SD of three replicate cultures), with *P*-values calculated using a two-tailed unpaired *t* test. **(C)** Samples of blood plasma were collected at the indicated times from mice transplanted with CBMCs alone or with iNKT cells, and analyzed for human IFN-γ by ELISA. Symbols represent mean ± SEM from 8–14 individual mice; data aggregated from four independent experiments. **Indicates *P* = 0.0033 as determined by two-tailed Mann–Whitney test. **(D)** Cord blood T cells were labeled with CTV dye and co-cultured with autologous monocytes alone, or with monocytes and iNKT cells, in medium containing IL-7 and IL-2 to drive T-cell expansion. Left plot shows expansion index of the cord T cells after 3–7 d, as determined by flow cytometric analysis of CTV staining. Middle and right plots show percentage of cord T cells staining positively for TNF-α (middle) or IFN-γ (right) after PMA and ionomycin stimulation. *P*-values calculated by two-tailed nonparametric paired *t* test. **(E)** NSG mice were transplanted with isolated cord T cells and autologous monocytes in the presence or absence of allogeneic iNKT cells. Samples of blood plasma were collected at the indicated times posttransplantation and analyzed for human IFN-γ by ELISA; symbols represent mean ± SEM from 5 mice tested in parallel.

and monocytes were required for the inhibition of cord T-cell IFN-γ production, or whether secreted factors were sufficient. Purified cord T cells were placed in wells coated with anti-CD3 and anti-CD28 antibodies in the presence or absence of transwell inserts containing autologous monocytes and allogeneic iNKT cells for 3–7 d. Intracellular cytokine staining after PMA and ionomycin stimulation of the

cord T cells revealed a significant reduction in the percent that produced IFN-γ in cultures that were exposed to transwells containing iNKT cells and monocytes (Fig 7A). These results demonstrated that soluble factors produced during iNKT-monocyte interactions were sufficient to inhibit cord T cells from acquiring IFN-γ effector function in response to TCR stimulation.

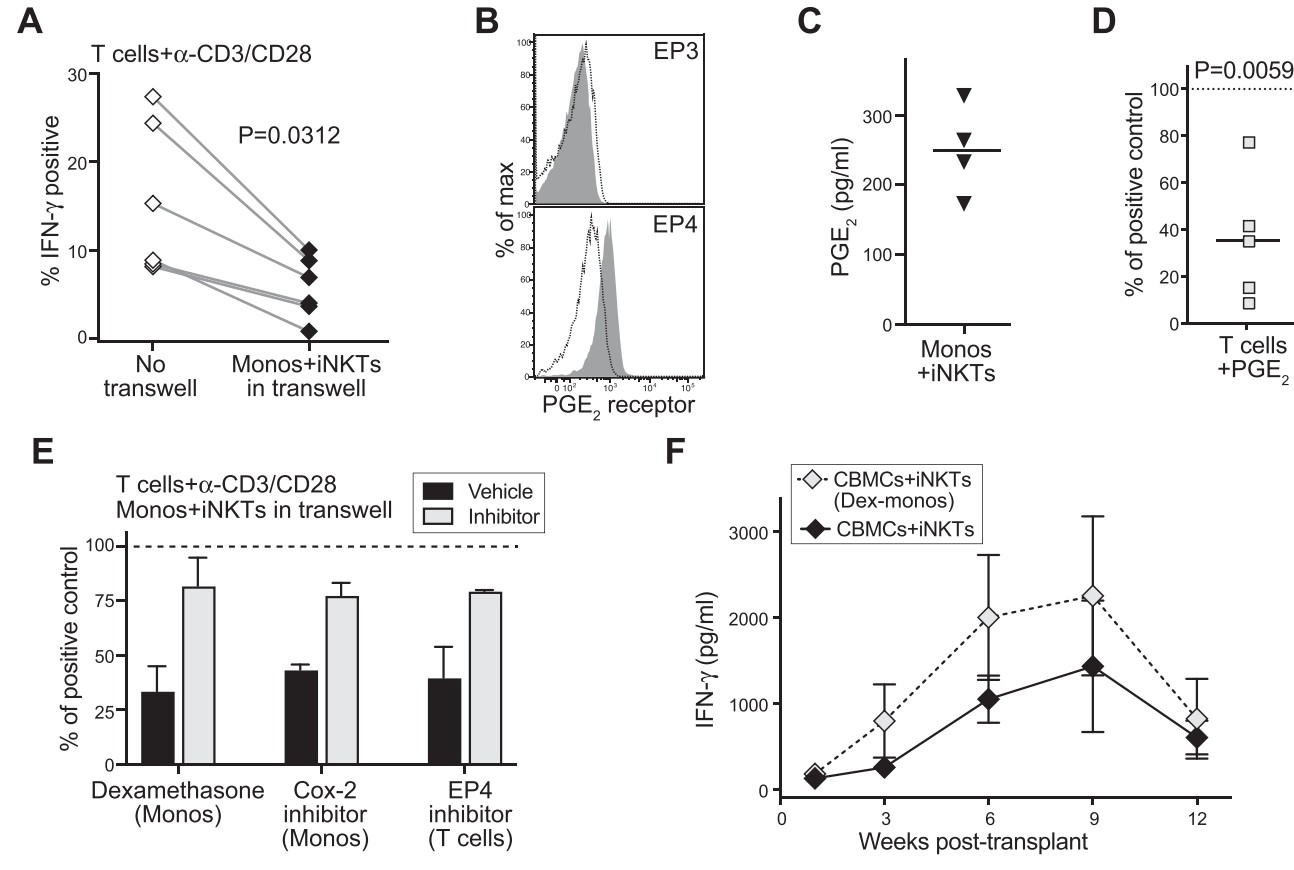

**Figure 7. Eicosanoids produced during invariant natural killer T (iNKT)–monocyte interactions suppress cord T cells.**
**(A)** Isolated CD4+ cord blood T cells were cultured for 3–5 d with anti-CD3 and anti-CD28 antibodies in the presence or absence of transwell inserts containing iNKT cells and cord blood monocytes. Plot shows percentage of T cells staining positively for intracellular IFN-γ after PMA/ionomycin stimulation; paired symbols show results from independent experiments. *P*-value calculated by two-tailed nonparametric paired *t* test. **(B)** Flow cytometric staining of cord T cells for PGE₂ receptors EP3 and EP4. Filled histograms show staining by specific antibodies; dotted line shows isotype control antibody. **(C)** iNKT cells and cord blood monocytes were co-cultured for 24 h and PGE₂ in the supernatant was quantitated using an enzyme assay. Symbols represent means from independent experiments. **(D)** Cord blood T cells were cultured for 3–4 d with anti-CD3 and anti-CD28 antibodies in medium containing 500 pg/ml PGE₂, then stimulated with PMA/ionomycin. Plot shows the IFN-γ+ T cells as a percent of the response by parallel cultures of cord T cells that were not exposed to PGE₂. Symbols show results from independent experiments. *P*-value calculated by two-tailed one sample *t* test. **(E)** Cord blood T cells were cultured for 3–5 d with anti-CD3 and anti-CD28 antibodies in the presence of transwell inserts containing iNKT cells and cord blood monocytes, then stimulated with PMA/ionomycin. Before co-culture, the monocytes were pretreated with dexamethasone (500 ng/ml) or Cox-2 inhibitor (NS-398, 10 μM), or the cord T cells were pretreated with an inhibitor of EP4 (L-161,982, 10 μM). Plot shows IFN-γ production from the transwell co-cultures as a percent of the response by positive control T cells that were cultured with anti-CD3/CD28 alone (dashed line). Bars represent mean ± SEM of two to three independent experiments. **(F)** NSG mice were transplanted with cord blood mononuclear cells and iNKT cells and blood samples were removed at the indicated times and tested for human IFN-γ by ELISA. Where indicated, the monocytes were isolated and treated with 500 pg/ml dexamethasone, then washed and added back to the cord blood mononuclear cells before injection. Symbols represent mean ± SEM from five mice tested in parallel.

We next sought to determine the nature of the inhibitory factor. Because experiments to block the release of proteins through the secretory pathway using brefeldin A pretreatment of the monocytes did not prevent the inhibitory effect (Fig S3), we investigated whether the relevant soluble factor might be an eicosanoid, such as prostaglandin E₂ (PGE₂). A prior study found that human cord blood T cells mainly express the EP4 receptor, which is a G-protein–coupled receptor that binds PGE₂ with high affinity and leads to cyclic AMP production (Li et al, 2014). Because the accumulation of cyclic AMP in T cells is suppressive (Arumugham & Baldari, 2017), this finding suggested that exposure to PGE₂ might inhibit cord T cells. Flow cytometric analysis for the two high affinity PGE₂ receptor subtypes, EP3 and EP4, demonstrated positive staining only for EP4 on our cord T cells (Fig 7B). Using an enzyme assay to test supernatants from iNKT-monocyte co-cultures for the presence of

PGE₂, we consistently observed detectable levels of PGE₂ (Fig 7C). Mass spectrometric analysis of supernatants from iNKT-monocyte co-cultures also revealed the presence of PGE₂ and associated products from its biosynthetic pathway (Fig S4A and B). Moreover, addition of synthetic PGE₂ to cord CD4+ T cells resulted in a significant reduction in their ability to produce IFN-γ after CD3/CD28 stimulation (Fig 7D). Thus, PGE₂ is released during iNKT-monocyte interactions, and exposure to PGE₂ inhibits the ability of cord T cells to produce IFN-γ in response to TCR stimulation.

We next investigated whether interrupting the PGE₂ pathway prevented the iNKT-monocyte inhibitory effects on cord T cells. Purified cord CD4+ T cells were stimulated by anti-CD3 and anti-CD28 in the presence of transwell inserts containing iNKT cells and monocytes, with the monocytes pretreated with dexamethasone, or a selective cyclooxygenase-2 (Cox-2) inhibitor, or with vehicle.

Similar to what we observed previously (Fig 7A), cord T-cell production of IFN-γ in response to anti-CD3/CD28 was inhibited when iNKT cells and vehicle-treated monocytes were present in the upper transwell (Fig 7E). However, cord T-cell IFN-γ production was nearly completely recovered when the monocytes in the upper well were pretreated with dexamethasone or Cox-2 inhibitor (Fig 7E). Moreover, pretreating the cord T cells with a selective inhibitor of the EP4 receptor also led to marked recovery of their IFN-γ production (Fig 7E). Thus, blocking eicosanoid production by the monocytes, or blocking the main receptor for PGE$_2$ on cord T cells, prevented the inhibition mediated by iNKT-monocyte soluble factors.

To confirm that similar iNKT-monocyte interactions are important for the suppression of cord T-cell responses in vivo, we tested the impact of pretreating monocytes with dexamethasone in vivo. Nonconditioned NSG mice were co-transplanted with iNKT cells and unmanipulated CBMCs, or with iNKT cells and CBMCs where the monocytes were first removed and pretreated with dexamethasone and then added back to the mixture before transplantation. Mice that received iNKT cells and CBMCs with dexamethasone pretreated monocytes showed a trend towards higher plasma IFN-γ levels than those that received iNKT cells and untreated CBMCs (Fig 7F), indicating that the inhibition of T-cell IFN-γ associated with co-transplantation of iNKT cells requires intact eicosanoid production by cord monocytes. Thus, in addition to production of the pro-hematopoietic cytokines IL-3 and GM-CSF, the nexus of iNKT cells, cord monocytes, and cord T cells was concomitantly associated with reduced T-cell secretion of inflammatory cytokines that adversely affect cord HSPC function.

## Discussion

The results presented here identify a powerful functionality of human iNKT cells that enables successful human hematopoietic engraftment in nonconditioned NSG mice. Prior studies have shown that NSG mice can serve as excellent hosts for human hematopoietic transplantation because they are deficient in immune cell types that ordinarily reject xenogeneic transplants (Shultz et al, 1995, 2005). However, conditioning was thought to be required to eliminate endogenous murine hematopoietic cells that would otherwise compete with transplanted human HSPCs for critical hematopoietic factors. In contrast, successful engraftment by human HSPCs can be achieved without conditioning in mouse strains that carry hypomorphic variants of the *c-kit* gene or in wild-type mice that have been pretreated with antibodies to block the KIT receptor (CD117) (Cosgun et al, 2014; McIntosh et al, 2015; Czechowicz et al, 2019). Thus, eradication of CD117[+] host cells or their genetic impairment is known to create a murine environment that is permissive for engraftment of human HSPCs without conditioning. However, it is a highly surprising observation that co-transplanting human iNKT cells with CBMCs appears to phenocopy this situation in NSG mice, a KIT-sufficient strain.

A key mechanism underlying this effect of the co-transplanted iNKT cells is probably that they supply critical hematopoietic factors that promote the engraftment of human HSPCs. It has previously been observed that iNKT cell–deficient mice show impaired hematopoiesis,

suggesting iNKT cells play a constitutive role in promoting hematopoietic activity in vivo (Kotsianidis et al, 2006). Moreover, prior studies have established that after administration of the highly potent synthetic antigen α-GalCer, murine iNKT cells produce GM-CSF and IL-3 in vivo, which leads to markedly enhanced colony-forming activity (Leite-de-Moraes et al, 2002; Kotsianidis et al, 2006). Consistent with this, we found that our human iNKT cells efficiently produced IL-3 and GM-CSF in response to CD1d-mediated presentation of α-GalCer, and they also produced these cytokines during interactions with cord blood monocytes and autologous T cells. Thus, it is likely that the co-transplanted iNKT cells promote cord blood HSPC activity in this model at least in part through secretion of GM-CSF and IL-3. In this way, they may produce a milieu in the host similar to that of knock-in mouse strains expressing human GM-CSF and IL-3, which have been shown to support improved multi-lineage human engraftment (Willinger et al, 2011; Rongvaux et al, 2014; Jangalwe et al, 2016).

However, pretransplant conditioning is still required for efficient engraftment of human HSPCs in human cytokine knock-in mouse strains, suggesting that the presence of human hematopoietic cytokines is not sufficient to allow transplanted human HSPCs to compete successfully with endogenous murine hematopoietic cells. Therefore, production of human GM-CSF and IL-3 probably does not fully explain the effects of co-transplanting iNKT cells and CBMCs in this model. Interestingly, we found that even without co-transplanted iNKT cells, transplanting purified cord blood HSPCs with autologous monocytes resulted in one to two orders of magnitude greater engraftment than transplanting purified HSPCs alone (compare Fig 4A and D). This suggests that cord blood monocytes support the engraftment of human HSPCs in nonconditioned NSG mice. We speculate that co-transplanting iNKT cells enhances this basal pro-hematopoietic effect of the cord blood monocytes. Specifically, given that PGE$_2$ has been identified as a powerful regulator of HSPC homing and survival that markedly enhances engraftment (North et al, 2007; Durand & Zon, 2010), we hypothesize that the ability of iNKT cells to induce PGE$_2$ secretion by cord monocytes (Figs 7C and S4A) is a key component of their engraftment promoting effects. Because eicosanoids such as PGE$_2$ are rapidly degraded in extracellular environments, this pathway would likely require co-localization of transplanted iNKT cells, monocytes, and HSPCs. It will thus be important to determine where and when such exposure might occur after transplantation.

Our analysis shows that PGE$_2$ is produced during interactions between iNKT cells and cord blood monocytes, without a requirement for added antigens. This is consistent with prior studies demonstrating that human iNKT cells recognize cellular lipids presented by CD1d molecules, including lyso-phosphatidylcholine (LPC) (Chang et al, 2008; Fox et al, 2009; Lopez-Sagaseta et al, 2012), which is constitutively expressed at low levels by many cell types and is also produced at high levels during eicosanoid biosynthesis (Exton, 1994; Funk, 2001). We show here that PGE$_2$ produced during iNKT interactions with cord monocytes inhibits cord T cells through the EP4 receptor, and that this pathway is active in vivo because pretreating the monocytes with dexamethasone resulted in increased circulating human IFN-γ in vivo after transplantation (Fig 7). This pathway likely also promotes HSPC engraftment because we observed that co-transplanting purified HSPCs with autologous monocytes and T cells resulted in engraftment failure, and this was

mitigated when iNKT cells were also co-transplanted (Fig 4D). Thus, we hypothesize that after transplantation cord blood T cells are activated to produce inflammatory cytokines and this response adversely affects autologous HSPCs, similar to effects observed previously in a model where total human UCB was transplanted into pre-conditioned NSG mice (Wang et al, 2017). Because exposure to iNKT cells and monocytes limited cord blood T-cell proliferation and TNF-α production in vitro (Fig 6D), and nearly completely abrogated their IFN-γ production (Fig 6D and E), it seems likely that the iNKT-monocyte regulatory axis promotes HSPC engraftment by limiting the adverse impact of cord T-cell inflammatory responses. However, in an unexpected and paradoxical twist, we also observed that secretion of factors that enhance HSPC activation was elevated when iNKT cells were co-incubated with both monocytes and T cells compared with co-incubation with either cell type alone (Fig 5C and E). Thus, our results point to a unique interrelationship, where iNKT cells and monocytes interact to limit the activation of cord T cells yet the presence of the T cells also enhances the release of pro-hematopoietic factors.

Interestingly, we observed little evidence of GVHD during these studies. About 5% of all the mice included in our analysis showed one or more signs of GVHD (weight loss, ruffled fur, jaundice, and eye irritation), and only one of these was a mouse that received iNKT cells. This lack of GVHD is consistent with observations that UCB transplantation in clinical settings is associated with comparatively low incidence of GVHD (Ballen et al, 2013; Keating et al, 2019), but contrasts with prior studies that have observed marked GVHD responses after transplantation of human cord blood cells into irradiated NSG mice (Wang et al, 2017). The relative lack of GVHD in our analysis likely results from minimizing the introduction of molecular patterns (e.g., conditioning-associated damage ligands, human red blood cell glycans) that have been shown to play a key role in the initiation of GVHD responses and xenograft rejection (Navarro-Alvarez & Yang, 2011; Ramadan & Paczesny, 2015; Toubai et al, 2016), but may also be due to tolerogenic effects of the iNKT cells because prior studies in murine models have shown that iNKT cells protect against GVHD by interacting with myeloid-derived suppressor cells to expand donor-derived regulatory T cells (Schneidawind et al, 2014, 2015; Du et al, 2017). Consistent with this, we observed that nearly half of the small population of human T cells present in the bone marrow of mice that received CBMCs + iNKT cells retained a naive phenotype (CD45RA[+]), whereas the T cells from mice that received CBMCs alone were nearly all CD45RO[+] (Fig 2D).

It is not clear whether co-transplantation of iNKT cells specifically promotes the establishment of the human B-cell compartment in our model. Murine iNKT cells have been shown to induce the formation of early germinal centers in a Bcl-6 dependent manner (Chang et al, 2011), and therefore it is possible that the presence of co-transplanted iNKT cells may contribute to the durable and productive B-cell compartment we observed in engrafted mice (Fig 3C–F). However, although we also noted that NSG mice transplanted with CBMCs and iNKT cells had a higher B: myeloid ratio compared to NBSGW mice transplanted with purified HSPCs (Fig 3A), it is possible that the KIT-sufficient status of the NSG host strain results in less efficient human myeloid differentiation in this strain.

Together, these findings support the potential utility of adult allogeneic iNKT cells as a cellular immunotherapy to improve outcomes of allogeneic human UCB transplantation. UCB transplantation in clinical settings is limited by high rates of graft failure and by longer times to neutrophil and platelet recovery compared with bone marrow and G-CSF mobilized peripheral blood transplants (Brown & Boussiotis, 2008; Lucchini et al, 2015). Our results suggest that iNKT-mediated production of hematopoietic factors and silencing of T-cell inflammation produces an environment that is favorable for cord blood HSPC engraftment, and thus might significantly improve these limitations. However, further studies will be needed to determine the impact of co-transplanted iNKT cells on anti-microbial immunity and GVT activity after transplantation because it is possible that the iNKT-monocyte axis described here may suppress immune responses that provide protection against infections and cancer relapse. In this regard it is important that we have previously found that eicosanoid-producing interactions between human iNKT cells and monocyte-derived DCs actually led to enhanced neutrophil-mediated control of cutaneously administered *Candida albicans* (Xu et al, 2016). Moreover, increased iNKT cell frequency after human haplo-identical hematopoietic transplantation is associated with lower cancer relapse rates (de Lalla et al, 2011; Casorati et al, 2012), and human iNKT cells expanded from blood or donor lymphocyte infusions have been shown to directly lyse primary human leukemic blasts in vitro (Schmid et al, 2018; Jahnke et al, 2019). Thus, there is reason for optimism that iNKT cell immunotherapy may actually promote key aspects of immune function after hematopoietic transplantation, while also limiting the inflammatory activation of cord T cells and promoting HSPC engraftment.

# Materials and Methods

### Cord blood

De-identified human UCB samples were acquired from University of Colorado's ClinImmune Labs cord blood bank, or from the Medical College of Wisconsin's tissue bank, or from the Obstetrics and Gynecology department at Meriter Hospital in Madison, WI. Blood samples were diluted with a 1:1 volume of leukocyte isolation buffer (PBS containing 2% bovine calf serum and 1 mM EDTA), and mononuclear cells were isolated by density gradient centrifugation using Ficoll-paque PLUS (GE Healthcare). If red blood cell contamination was apparent, the isolated cells were resuspended in leukocyte isolation buffer and the density gradient purification was repeated. For some experiments CBMCs were specifically depleted of T cells, or cells populations of interest (e.g., cord T cells or monocytes) were isolated using RosetteSep reagents from StemCell Technologies, or by magnetic sorting using reagents from Miltenyi Corp.

### iNKT cells

Human blood samples were collected and used in accordance with University of Wisconsin-Madison IRB protocol 2018-0304. Human CD4[+] iNKT cells were sorted from peripheral blood of healthy adult

volunteer donors using human CD1d tetramers loaded with synthetic lipid antigen, and expanded in vitro using irradiated PBMC feeder cells, phytohemagglutinen, and recombinant human IL-2. iNKT cells used in these analyses included CD4[+] clonal lines that we have previously established (Brigl et al, 2006; Chen et al, 2007; Fox et al, 2009), and short-term polyclonal cultures of CD4[+] iNKT cells sorted using α-GalCer loaded CD1d tetramer provided by the NIH tetramer facility at Emory University (Sharma et al, 2018). Clonal and polyclonal iNKT cell cultures were maintained in culture medium ("T cell medium") comprised RPMI 1640 medium diluted with 10% heat-inactivated fetal bovine serum (GeminiBio), 5% heat-inactivated defined/supplemented bovine calf serum (Hyclone), 3% pooled human AB serum (Atlanta Biologicals), 1% L-glutamine (2 mM final concentration), 1% penicillin/streptomycin (100 IU/ml and 100 μg/ml final concentration, respectively), and 200 U/ml recombinant human IL-2 (Peprotech). All iNKT cultures used for experiments were of 98–100% purity as assessed by flow cytometric analysis with α-GalCer loaded CD1d tetramers.

## Transplantation of human cells into mice

Animal studies using NSG (NOD.Cg-Prkdc[scid] Il2rg[tm1Wjl]/SzJ) and NBSGW (NOD.Cg-Kit[W-41J] Tyr[+] Prkdc[scid] Il2rg[tm1Wjl]/ThomJ) mice were performed in accordance with UW-Madison IACUC protocol M005199. Mice were maintained using autoclaved cages, bedding, and food. Because preliminary studies revealed that male NSG mice did not support successful engraftment (see Fig S5), only female NSG mice were included in this study. Human cells were resuspended in 150–200 μl sterile PBS per mouse and injected intravenously (retro-orbital route) under isoflurane anesthesia. NSG mice received 5 × 10[6] CBMCs alone, or in combination with 0.5–1 × 10[6] allogeneic iNKT cells. Alternatively, where indicated, NSG mice were transplanted with 0.5–1 × 10[5] purified CD34[+] cells alone or in combination with one or more of the following: 0.5–1 × 10[5] purified autologous monocytes, 2 × 10[6] purified autologous T cells, and 0.5 × 10[6] allogeneic iNKT cells. Mice were randomly assigned to treatment groups, ear-tagged, and co-housed in cages with mice from other treatment groups. Mice were weighed and monitored for signs of GVHD (ruffled fur, hunching, squinting, lethargy) weekly, and those showing signs of GVHD were excluded from the analysis. Mice were euthanized by $CO_2$ for analysis of tissues at prospectively determined time points (day 100 posttransplant, unless otherwise indicated).

## Flow cytometric analysis

Bone marrow was collected from one femur of each mouse by removing one end of the bone and centrifuging at 8,000$g$ for 60 s. Spleen cells were prepared by gently homogenizing the tissue using a blunt plastic syringe. Cell samples were resuspended in flow cytometry buffer (PBS containing 10% pooled human AB serum) and filtered through a 40 μM nylon mesh before staining with fluorescently conjugated antibodies. Monoclonal antibodies used for staining were purchased from BioLegend and included the following antibody clones: CD3 (OKT3), CD14 (M5E2), CD19 (HIB19), CD34 (8G12), CD38 (HIT2), human CD45 (HI30), murine CD45.1 (A20), CD123 (6H6), human IFN-γ (4S.B3), human TNF-α (MAb11), and pan-HLA

Class I (W6/32). Cell number was quantified using Precision Count Beads (BioLegend).

## Immunohistochemical analysis

Formalin-fixed, paraffin-embedded tissue sections were deparaffinized and hydrated, and endogenous peroxidase activity was blocked with 0.3% hydrogen peroxidase solution, and nonspecific labeling was blocked in a 5% goat serum blocking solution. Two-color staining was performed using VectorLabs ImmPRESS Duet Double Staining System with a rabbit anti-human CD20 polyclonal antibody (Thermo Fisher Scientific) and a mouse monoclonal antibody against Bcl-6 (clone PGB6P; Santa Cruz Biotechnology). Serial sections of the same tissue were stained for human IgM or IgG using specific rabbit polyclonal antibodies (Cell Marque) with the Super Sensitive Polymer-Horseradish Peroxidase Immunohistochemistry detection system (BioGenex) using diaminobenzidine color development. Sections were lightly counter-stained with hematoxylin to visualize cell nuclei.

## Live cell imaging of interactions between iNKT cells and CBMC populations

T cells, B cells, and monocytes were isolated from a CBMC sample and stained with IncuCyte Cytolight Rapid Green reagent (Sartorius). CD4[+] iNKTs were stained with IncuCyte Cytolight Rapid Red reagent (Sartorius). iNKT cells (1 × 10[5] cells per well) placed into replicate wells of a 48-well plate with cord T cells, B cells, or monocytes (2 × 10[4] cells per well) in 1 ml of media (RPMI, 10% BCS, 3% human serum, non-essential amino acids, and pen/strep). Cellular interactions were analyzed over a 24-h period in an IncuCyte S3 Live-Cell Analysis system, with nine images per well taken every 15 min with a 20× objective lens. Images were analyzed using IncuCytes analysis software for both the number and total area of overlap between iNKT cells and the indicated cell population.

## In vitro analysis of hematopoietic expansion

CD34[+] cells (HSPCs) were purified from CBMCs by magnetic sorting using positive selection. To assess the impact of secreted factors on HSPC differentiation, 2.5 × 104 HSPCs were seeded into the lower well of 24-well plates, and iNKT cells, cord monocytes, and/or cord T cells (1.5 × 10[4] cells each) were added to transwell inserts. Where indicated, a 1:1 ratio of CD3/CD28 Dynabeads (Thermo Fisher Scientific) was added to activate the iNKT cells. The cells were cultured at 37°C and 5% $CO_2$ in X-VIVO15 serum-free medium (Lonza) containing 20 ng/ml each of Flt3-L, TPO, SCF, and IL-7 (Peprotech). After 10 d, cells from the lower well were stained with fluorescently labeled antibodies against CD45RA (Clone: HI100), CD33 (Clone: WM53), CD38 (Clone: HB-7), and CD34 (Clone: 581) and analyzed by flow cytometry using Precision Count Beads (BioLegend) to determine the number of cells bearing markers of differentiation. To assess the impact of exposure to T cellcytokines, CD34[+] HSPCs were isolated from an HLA-A2[+] CBMC sample. The HSPCs (1 × 10[4] cells) alone or in combination with 1 × 10[5] CD34-depleted CBMCs (HLA-A2 negative sample) were placed into replicated wells of a 96-well

plate. Where indicated, stimulating antibodies (anti-CD3 and anti-CD28) or blocking antibody (anti-IFN-γ or anti-TNF-α) were added to a final concentration of 5 μg/ml. After 7 d of culture, the cells were resuspended and HLA-A2⁺ cells were quantitated by flow cytometry.

### Determination of iNKT cell production of GM-CSF and IL-3

iNKT cell cytokine production in response to CD1d molecules presenting titrated doses of a lipid antigen was assessed as previously described (Brigl et al, 2006; Wang et al, 2008). Briefly, high protein binding 96-well plates were coated with 10 μg/ml recombinant CD1d-Fc fusion protein. α-GalCer lipid antigen suspended in DMSO was sonicated in a heated water bath for 45 min, then diluted in PBS and added to the wells at the indicated concentrations and incubated for 24 h at 37°C. The plates were washed first with PBS, then with T-cell medium lacking IL-2. Polyclonal iNKT cells suspended in T-cell medium lacking IL-2 were added at $5 \times 10^4$ cells per well, and incubated at 37°C and 5% $CO_2$. Culture supernatants were harvested after 24 h and concentrations of IL-3 and GM-CSF were determined by ELISA. Where indicated, monocytes and T cells were isolated from the same cord blood sample by magnetic sorting and co-cultured at a 1:1 ratio with allogeneic iNKT cells for 48 h, and concentrations of IL-3 and GM-CSF in culture supernatants were determined by ELISA.

### Quantitation of human IFN-γ in murine plasma

Blood was collected by retro-orbital bleeding from isoflurane anesthetized mice at the indicated time points posttransplantation. Blood samples were diluted with a 1:1 ratio of PBS, then centrifuged at 400*g* for 4 min to pellet the red blood cells, and the supernatant was collected. Supernatants were diluted 1:5 in ELISA assay buffer (PBS with 1 mg/ml BSA), and human IFN-γ was detected from three replicate wells using an antibody pair (MD-1 and 4S.B3) purchased from BioLegend. IFN-γ concentration was determined by interpolation from a standard curve of recombinant human IFN-γ (Peprotech) that was tested in parallel.

### In vitro analysis of NKT-monocyte suppression of cord T cells

Monocytes were positively selected from freshly isolated CBMCs using CD14 magnetic beads, then untouched CD4⁺ T cells were isolated by magnetically depleting other subsets (Miltenyi Biotec). Cord T cells were labeled with 5 μM Cell Trace Violet (CTV) dye (Thermo Fisher Scientific). Cord T cells ($5 \times 10^4$/well) were cultured with autologous monocytes ($2.5 \times 10^4$ well) in the presence or absence of allogeneic iNKT cells ($0.5 \times 10^4$/well) in culture medium (RPMI, 15% BCS, 3% human serum, Pen/Strep) containing 50 U/ml recombinant human IL-2 and 1 ng/ml recombinant human IL-7 (Peprotech) in a humidified incubator at 37°C with 5% $CO_2$. After 3–7 d of co-culture, the cells were stained with antibodies against CD3, CD14, and iNKT TCR, and after gating out iNKT cells and monocytes proliferation of the cord T cells of was assessed by flow cytometry to determine dilution CTV fluorescence intensity. Alternatively, the cells were stimulated with PMA/ionomycin in the presence of brefeldin A (BioLegend) for 4–6 h, then fixed and permeabilized and stained intracellularly for IFN-γ and TNF-α. To assess the impact of

secreted factors produced by iNKT cells and monocytes, cord T cells were placed into the lower wells of transwell plates with anti-CD3 and anti-CD28 antibodies, in the presence or absence of transwell inserts containing monocytes and iNKT cells ($5 \times 10^4$ cells of each). Where indicated, the specified cell populations (monocytes or T cells) were treated with inhibitor drugs or vehicle for 90 min at 37°C then washed twice before being added to the wells.

### Analysis of secreted $PGE_2$

Monocytes were positively selected from freshly isolated CBMCs using CD14 magnetic beads. Cultured iNKT cells were placed in medium lacking IL-2 for 24 h before the experiment to ensure they were in a resting state. Monocytes ($1 \times 10^6$ cells) and iNKT cells ($1 \times 10^6$ cells) were combined in 1 ml of culture medium (RPMI containing 15% BCS, 3% pooled human AB serum, 1% L-glutamine, and 1% Penicillin/Streptomycin). After 24 h of co-incubation at 37°C with 5% $CO_2$, culture supernatants were harvested and $PGE_2$ levels were quantitated in duplicate using the DetectX $PGE_2$ Multi-Format ELISA kit (Arbor Assays) following the manufacturer's protocol. For mass spectrometric determination of eicosanoids, cell-free culture supernatants were collected immediately after combining monocytes and iNKT cells ("Time 0") and after 24 h of co-culture. The internal standard deuterated-$PGE_2$ ($PGE_2$-d4; Cayman Chemical) was added to each sample and incubated for 30 min on ice before extraction with two volumes of ice cold methanol. Samples were concentrated to 1 ml by $N_2$ gas evaporation and then rapidly acidified with 9 ml of pH 3.5 $H_2O$ before loading onto C18 columns (Isolute; Biotage) that were previously conditioned with 6 ml MeOH followed by 6 ml neutral $H_2O$. Sample tubes were washed with 4 ml neutral water and loaded onto the same C18 column. The columns were then washed with 3 ml hexane and eluted using 6 ml methyl formate followed by 2 ml MeOH and concentrated to ~50 μl by $N_2$ gas evaporation. Samples were diluted to a final volume of 200 μl using 55:45 MeOH:$H_2O$, and loaded onto an HPLC coupled to a mass spectrometer (Q Exactive; Thermo Fisher Scientific) using a C18 Acquity BEH column (100 × 2.1 mm × 1.7 μm) operated in negative ionization mode. Samples were eluted using a gradient of MeOH:$H_2O$:CH3COOH solvent that transitioned from 55:45:0.1 to 98:2:0.1 over a course of 28 min. Scans were collected from 3.5 to 20 min for mass-to-charge ratios (m/z) of 100–800 and analyzed using Maven open source software (omicX) in comparison to standards (Cayman Chemical) normalized to relative amounts of $PGE_2$-d4.

## Supplementary Information

## Acknowledgements

NJ Hess supported by National Institutes of Health (NIH) T32 AI125231 and NIH T32 HL07899. EA Bobeck supported by NIH T32 HL07899. Project support provided by NIH R21 AI116007 and R01 AI136500 to JE Gumperz, and R21 AI105841 (AW Hudson).

## Author Contributions

NJ Hess: conceptualization, data curation, formal analysis, investigation, methodology, and writing—review and editing.
NS Bharadwaj: conceptualization, data curation, formal analysis, investigation, methodology, and writing—review and editing.
EA Bobeck: conceptualization, data curation, formal analysis, investigation, and methodology.
C McDougal: investigation and methodology.
S Ma: investigation and methodology.
J-D Sauer: resources, supervision, and validation.
AW Hudson: resources.
JE Gumperz: conceptualization, resources, data curation, formal analysis, supervision, funding acquisition, visualization, methodology, project administration, and writing—original draft, review, and editing.

## Conflict of Interest Statement

The authors declare that they have no conflict of interest.

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
