## [Reviewer comments · Life Science Alliance]

Life Science Alliance

iNKT cells coordinate immune pathways to enable human immune engraftment in non-conditioned hosts

Nicholas Hess, Nikhila Sham Sunder Bharadwaj, Elizabeth Bobeck, Courtney McDougal, Shidong Ma, John-Demian Sauer, Amy Hudson, and Jenny Gumperz

DOI: <https://doi.org/10.26508/lsa.202000999>

Corresponding author(s): Jenny Gumperz, University of Wisconsin School of Medicine and Public Health

Review Timeline:

Submission Date:	2020-12-17
Editorial Decision:	2021-01-29
Revision Received:	2021-03-31
Editorial Decision:	2021-05-27
Revision Received:	2021-05-27
Accepted:	2021-05-28

Scientific Editor: Shachi Bhatt

Transaction Report:

January 29, 2021

Re: Life Science Alliance manuscript #LSA-2020-00999-T

Dr. Jenny E Gumperz
University of Wisconsin School of Medicine and Public Health
Department of Medical Microbiology and Immunology
Microbial Sciences Building
1550 Linden Dr.
Madison, WI 53706

Dear Dr. Gumperz,

Thank you for submitting your manuscript entitled "iNKT cells orchestrate a pro-hematopoietic switch that enables human immune engraftment in non-conditioned hosts" to Life Science Alliance. The manuscript was assessed by expert reviewers, whose comments are appended to this letter.

As you will note from the accompanying reviews, both reviewers seem intrigued by these findings but have also raised some important questions that should be addressed prior to further consideration of the manuscript at LSA. We, thus, encourage you to submit a revised version of the manuscript to LSA that addresses all of the reviewers' points. While we agree with Reviewer 2 that figuring out whether autologous iNKTs would work better is an interesting and important question (R2 pt 3), it would not be required for you to show these data experimentally in the revised manuscript for LSA, a discussion should suffice.

Thank you for this interesting contribution to Life Science Alliance. We are looking forward to receiving your revised manuscript.

Sincerely,

Shachi Bhatt, Ph.D.
Executive Editor
Life Science Alliance
<https://www.lsjournal.org/>
Tweet @SciBhatt @LSAJournal

- A letter addressing the reviewers' comments point by point.
- An editable version of the final text (.DOC or .DOCX) is needed for copyediting (no PDFs).
- High-resolution figure, supplementary figure and video files uploaded as individual files: See our detailed guidelines for preparing your production-ready images, <https://www.life-science-alliance.org/authors>
- Summary blurb (enter in submission system): A short text summarizing in a single sentence the study (max. 200 characters including spaces). This text is used in conjunction with the titles of papers, hence should be informative and complementary to the title and running title. It should describe the context and significance of the findings for a general readership; it should be written in the present tense and refer to the work in the third person. Author names should not be mentioned.

B. MANUSCRIPT ORGANIZATION AND FORMATTING:

Reviewer #1 (Comments to the Authors (Required)):

Hess, Gumperz et al. present an interesting manuscript that expands on the existing literature that iNKT can suppress GvHD and enhance anti-tumor activity of HSCTs (mostly well cited refs., there are some looking at iNKT subsets that suppress GvHD and promote anti-tumor responses) as well as work from this group on iNKT interactions with various immune and hemopoietic cell types. They

show that iNKT can promote multi-lineage engraftment in classic hu-mouse NSG model. The focus is Umbilical Cord Blood transplants, an attractive approach that hasn't been fully realized in the clinic due to limitations noted.

A well-performed and written study with experimental and clinical implications. Modest revisions would significantly improve the MS. This should then be of value to groups wanting to use such models more readily and widely with minimal conditioning, as well as the potential relevance to human transplants. Mechanisms are reported to include direct interaction with (CD1d+) monocytes and (more commonly CD1d-negative) T cells, leading to IL-3 as well as GM-CSF secretion and a novel role for PGE2. (CD1d+) B cells and serum Igs also appeared about the same time as clinically post transplant. It is also interesting that the iNKT did not persist long after transfer despite the profound and long term effect on engraftment.

Why only iNKT cells were considered to aid engraftment is not immediately clear, but is not a major limitation given the interesting and comprehensively studied results.

Results:

"Pre-transplant conditioning is typically required for successful engraftment of purified human HSPCs transplanted into immunodeficient murine hosts." Good to cite here.

"... transplanted CBMCs alone ... there was typically only a small population of human cells detected in the bone marrow after 3 months (Fig 1A, middle row). Moreover, the human population found in these mice showed little or no positive staining for CD34" what were these cells ?

It is interesting that T cells persisted and dominated in the in the bone marrow of NSG mice that received CBMCs alone, showing lack in the other mice groups was not related to thymus defects or lack of T cell "lineage+ cells" in the grafts.

"Plasma samples taken at 5-9 months post-transplantation typically contained clearly detectable levels of human immunoglobulin, confirming the presence of immunoglobulin-producing human B cells (Fig. 3F)." Can absolute amount of Igk (measure of Igs used) be calculated to compare to post-HSCT.

"Co-transplanting iNKT cells with purified HSPCs was not sufficient to promote successful engraftment". This is a significant result and could be mentioned in the Abstract.

"Notably, while CD1d staining appeared uniformly positive on cord blood monocytes, CD1d appeared only to be expressed at low levels on a fraction of cord T cells" but is present on normal B cells.

While the IFNg data is competing, iNKT are a major source of IFNg too. Please address.

"transplanting purified cord T cells with autologous monocytes was sufficient to produce a systemic burst of human IFN-g that peaked at 6 weeks after transplantation". I'm not sure that a gradual peak rising to 6 weeks is a "burst".

"To confirm that similar iNKT-monocyte interactions are important for the suppression of cord T cell responses in vivo, we tested the impact of pre-treating monocytes with dexamethasone in vivo." A more direct way would have been to employ CD1d antibodies in vitro (in the part model) or preferably in vivo which might impact engraftment and should last long enough to have an impact

given iNKT mostly gone by 5 weeks.

Reviewer #2 (Comments to the Authors (Required)):

Overall a well written manuscript with sound experimental design and interesting results with high potential for translation in clinic. I have following comments/suggestions (required) for the authors:

1. What happens when iNKTs are combined with conditioning/irradiation.. will it lead to even more improved engraftment in the mice?
2. In the figure 2C please characterize the T cells which seem to be mediating rejection (CD4/CD4/Tcon/scm/em/temra?)
3. Will autologous iNKTs work better (compared to third party/allogeneic used in the experiments in this study)?
4. Suggest assessing GM-CSF/IL-3 in the sera of mice which get CBM plus / minus iNKT
5. Similarly, will giving mice these factors (GM-CSF/IL-3) be enough to enhance HSC engraftment similar to iNKTs?
6. They nicely show effect of iNKTs on IFN γ and TNF α production in vitro, does this finding also help explain less GvHD seen with higher numbers of iNKTs early after allogeneic HSCT?
7. Suggest shortening introduction to focus on the problem addressed in this study.

Jenny E. Gumperz, Ph.D.
Professor of Med. Micro. & Immunol.
University of Wisconsin SMPH
1550 Linden Dr.
Madison, WI 53706
Tel. (608) 263-6902
email: jegumperz@wisc.edu

March 30th, 2021

Dear Editors,

Attached, please find the revised version of our manuscript entitled "iNKT cells orchestrate a pro-hematopoietic switch that enables engraftment in non-conditioned hosts" (LSA-2020-00999-T). We would like to sincerely thank both reviewers for providing thoughtful critiques that have allowed us to strengthen our manuscript. Based on these critiques, we have added new data (Figs 2D and 5D), revised Fig 3F, and made revisions to the manuscript text that are highlighted in yellow.

Our point-by-point response to the specific points raised by the Reviewers is as follows:

Reviewer 1

1. *"Pre-transplant conditioning is typically required for successful engraftment of purified human HSPCs transplanted into immunodeficient murine hosts." Good to cite here.*

We added citations of 3 references to support this statement.

2. *"... transplanted CBMCs alone ... there was typically only a small population of human cells detected in the bone marrow after 3 months (Fig 1A, middle row). Moreover, the human population found in these mice showed little or no positive staining for CD34" what were these cells ?*

We added text to clarify here that the human cells are T cells (as shown in Fig. 2C, middle panel). We also added new data (Fig. 2D) to better characterize the T cells found in NSG mice transplanted with CBMCs +/- iNKT cells.

3. *"Plasma samples taken at 5-9 months post-transplantation typically contained clearly detectable levels of human immunoglobulin, confirming the presence of immunoglobulin-producing human B cells (Fig. 3F)." Can absolute amount of Igk (measure of Igs used) be calculated to compare to post-HSCT.*

Fig. 3F has been edited to show the estimated µg/ml Igk.

4. *"Co-transplanting iNKT cells with purified HSPCs was not sufficient to promote successful engraftment". This is a significant result and could be mentioned in the Abstract.*

We agree that it is a significant observation that the effect of iNKT cells was not observed when

Department of Medical Microbiology & Immunology

they were co-transplanted with purified HSCs, however, due to word count limitations in the abstract we decided to focus instead on the finding that their effects were due to interactions with cord monocytes and T cells. Therefore, we revised the abstract to better clarify our focus on understanding the immunological nexus underlying the engraftment outcome observed here.

5. *"Notably, while CD1d staining appeared uniformly positive on cord blood monocytes, CD1d appeared only to be expressed at low levels on a fraction of cord T cells" but is present on normal B cells.*

The text has been revised to acknowledge that CD1d is also uniformly expressed on cord B cells, although we did not see evidence of sustained iNKT cell contact with cord B cells.

6. *While the IFN γ data is competing, iNKT are a major source of IFN γ too. Please address.*

It seems that the co-transplanted iNKT cells themselves don't produce much IFN- γ in this model, since we observed that their presence is associated with reduced levels of circulating IFN- γ , compared to mice that received CBMCs alone. Prior studies by our lab and others have established that iNKT cells only produce IFN- γ under conditions associated with inflammation (e.g. when they are exposed to IL-12 or elevated ICAM-1), or in the presence of strong antigenic stimulation (e.g. α -GalCer). We previously established that under non-inflammatory conditions recognition of self antigens on APCs preferentially activates human iNKT cells to produce GM-CSF and IL-13 and not IFN- γ (Wang et al., Blood 2008). We noted this prior finding in the Discussion section of this manuscript, along with discussion of our new observation that iNKT cells efficiently produce IL-3 in response to weak TCR stimulation.

7. *"transplanting purified cord T cells with autologous monocytes was sufficient to produce a systemic burst of human IFN-g that peaked at 6 weeks after transplantation". I'm not sure that a gradual peak rising to 6 weeks is a "burst".*

The text has been edited to re-phrase the description of this result in a more circumspect way.

8. *"To confirm that similar iNKT-monocyte interactions are important for the suppression of cord T cell responses in vivo, we tested the impact of pre-treating monocytes with dexamethasone in vivo." A more direct way would have been to employ CD1d antibodies in vitro (in the part model) or preferably in vivo which might impact engraftment and should last long enough to have an impact given iNKT mostly gone by 5 weeks.*

We agree that using anti-CD1d blocking antibodies would be a way to investigate the iNKT-monocyte interaction (although we have found in the past that interpreting results from this type of approach is more complicated than one would think, likely due to artifacts resulting from monocyte expression of Fc receptors).

However, in the analysis performed here we wished to specifically test the impact on T cells of blocking iNKT-mediated induction of monocyte production of PGE₂, without preventing other regulatory pathways that might potentially result from iNKT-monocyte interactions (e.g. IL-10, TGF- β). We feel that the approach we used provides a more specific demonstration that the induction of PGE₂ secretion plays a key role in the iNKT-monocyte regulatory axis observed here.

Reviewer 2

1. What happens when iNKTs are combined with conditioning/irradiation.. will it lead to even more improved engraftment in the mice?

This is a fascinating question, but we feel strongly that this issue is beyond the scope of the current study, and should be addressed in a future analysis. With the realization in the field that inflammation produced by conditioning has direct adverse consequences on outcomes of HSCT, there has been substantial focus on identifying less damaging conditioning regimens and in some cases (e.g. HSCT treatment of immunodeficiencies where pre-conditioning is not needed to reduce tumor burden) it is now avoided altogether. We feel that our current analysis provides an important framework for future investigation into how iNKT cellular immunotherapy may be integrated with clinical approaches designed to mitigate the adverse effects of conditioning.

2. In the figure 2C please characterize the T cells which seem to be mediating rejection (CD4/CD4/ Tcon/scm/em/temra)?

We have added new data (Fig. 2D) to better characterize the T cell populations in mice transplanted with CBMCs +/- iNKT cells.

3. Will autologous iNKTs work better (compared to third party/allogeneic used in the experiments in this study)?

Consistent with the almost complete lack of allelic polymorphism at the amino acid level in human CD1d molecules, we have consistently found no differences between iNKT cell responses to autologous and allogeneic APCs. It is possible that autologous iNKT cells might persist longer following transplantation, since they would not be susceptible to an allo-response by the cord T cells. However, from a practical standpoint, it is likely to be much more feasible to use allogeneic iNKT cells prepared from adult blood for this type of immunotherapy since these could be generated ahead of time and stored frozen, then simply co-administered with the cord cells at the time of transplant. (In contrast, generating immunotherapeutic iNKT cells that are autologous to the umbilical cord graft is likely to be problematic due to the need to first isolate the iNKT cells from the cord sample and to store the rest of the UCB cells while the iNKT cells are being expanded in vitro.) For this reason, we specifically chose to use allogeneic iNKT cells expanded from adult peripheral blood for these studies.

4. Suggest assessing GM-CSF/IL-3 in the sera of mice which get CBM plus / minus iNKT.

We performed the suggested experiment, and have added a new figure (5D) showing that both GM-CSF and IL-3 are clearly elevated in bone marrow of mice that got CBMCs+iNKTs compared to those that got CBMCs alone.

5. Similarly, will giving mice these factors (GM-CSF/IL-3) be enough to enhance HSC engraftment similar to iNKTs?

Interestingly, this experiment has essentially been done (by others) through the use of knock-in mouse strains expressing human GM-CSF and IL-3. As noted in the Discussion of our manuscript, these mice do support improved human multi-lineage engraftment, but they nevertheless appear to require pre-conditioning. Therefore, we speculate that co-transplanting iNKT cells likely brings an additional pre-hematopoietic factor into play, and that this may be their ability to induce monocyte secretion of PGE₂, since this has been shown by others to have

Department of Medical Microbiology & Immunology

far-reaching effects on HSPC survival and engraftment.

6. They nicely show effect of iNKTs on IFN γ and TNF α production in vitro, does this finding also help explain less GvHD seen with higher numbers of iNKTs early after allogeneic HSCT?

We suspect that the tolerogenic effects of the iNKT-monocyte axis identified here may indeed play a role in the link between iNKT cells and reduced GVHD, but in the interest of maintaining focus and conciseness we decided not to go into this question.

7. Suggest shortening introduction to focus on the problem addressed in this study.

The introduction has been revised to make it more focused and specific to this study.

In summary, we feel these revisions have addressed the concerns raised, and have substantially improved the manuscript. We thank the editors and reviewers for their interest and for their extremely valuable input, which has been tremendously helpful in further strengthening our work.

Sincerely,

Jenny Gumperz, Ph.D.

May 27, 2021

RE: Life Science Alliance Manuscript #LSA-2020-00999-TR

Dr. Jenny E Gumperz
University of Wisconsin School of Medicine and Public Health
Department of Medical Microbiology and Immunology
Microbial Sciences Building
1550 Linden Dr.
Madison, WI 53706

Dear Dr. Gumperz,

Thank you for submitting your revised manuscript entitled "iNKT cells coordinate immune pathways to enable human immune engraftment in non-conditioned hosts". We would be happy to publish your paper in Life Science Alliance pending final revisions necessary to meet our formatting guidelines.

Please also attend to the following:

- please make sure the author order in your manuscript and our system match
- please use the [10 author names, et al.] format in your references (i.e. limit the author names to the first 10)
- please add callouts for Figure S4A, B to your main manuscript text
- we encourage you to revise the legend of Figure S1 to be sure that it perfectly matches the actual figure (there is mention of panels A and B which are not marked in the actual figure)
- please add scale bars for Figure 3E

A. FINAL FILES:

-- High-resolution figure, supplementary figure and video files uploaded as individual files: See our detailed guidelines for preparing your production-ready images, <https://www.life-science->

alliance.org/authors

B. MANUSCRIPT ORGANIZATION AND FORMATTING:

Sincerely,

Shachi Bhatt, Ph.D.
Executive Editor
Life Science Alliance
<http://www.lsjournal.org>
Tweet @SciBhatt @LSAJournal

Reviewer #2 (Comments to the Authors (Required)):

The authors have addressed all my concerns.

May 28, 2021

RE: Life Science Alliance Manuscript #LSA-2020-00999-TRR

Dr. Jenny E Gumperz
University of Wisconsin School of Medicine and Public Health
Department of Medical Microbiology and Immunology
Microbial Sciences Building
1550 Linden Dr.
Madison, WI 53706

Dear Dr. Gumperz,

Thank you for submitting your Research Article entitled "iNKT cells coordinate immune pathways to enable human immune engraftment in non-conditioned hosts". It is a pleasure to let you know that your manuscript is now accepted for publication in Life Science Alliance. Congratulations on this interesting work.

DISTRIBUTION OF MATERIALS:

Again, congratulations on a very nice paper. I hope you found the review process to be constructive and are pleased with how the manuscript was handled editorially. We look forward to future exciting submissions from your lab.

Sincerely,

Shachi Bhatt, Ph.D.

Executive Editor

Life Science Alliance

<http://www.lsjournal.org>
